# SCD-MMPSR: Semi-Supervised Cross-Domain Learning Framework for Multitask Multimodal Psychological States Recognition

## Abstract

Modern human-computer interaction interfaces demand robust recognition of complex psychological states in real-world, unconstrained settings. However, existing multimodal corpora are typically limited to single tasks with narrow annotation scopes, hindering the development of general-purpose models capable of multitask learning and cross-domain adaptation. To address this, we introduce SCD-MMPSR (**S**emi-supervised **C**ross-**D**omain **M**ultitask **M**ultimodal **P**sychological **S**tates **R**ecognition), a novel framework that unifies heterogeneous corpora via GradNorm-based adaptive task weighting in multitask semi-supervised learning (SSL) to train models across diverse psychological prediction tasks jointly. At the architectural core, we propose two innovations within a graph-attention backbone: (1) Task-Specific Projectors, which transform shared multimodal representations into task-conditioned logits and re-embed them into a unified hidden space, enabling iterative refinement through graph message passing while preserving modality alignment; and (2) a Guide Bank, a learnable set of task-specific semantic prototypes that anchor predictions, injecting structured priors to stabilize training and enhance generalization. We evaluate SCD-MMPSR on three distinct psychological state recognition tasks, emotion recognition (MOSEI), personality trait recognition (FIv2), and ambivalence/hesitancy recognition (BAH), demonstrating consistent improvements in multitask performance and cross-domain robustness over strong baselines. We also evaluate the generalization of SCD-MMPSR on unseen data from the MELD dataset. Multitask SSL improves generalization on MELD by a macro F1-score of 7.5% (35.0 vs. 27.5) compared to single-task SSL. Our results highlight the potential of semi-supervised, cross-task representation learning for scalable affective computing. The code is available at https://github.com/Anonymous-user-2026/ICLR_2026.

## 1 Introduction

Effective human-computer interaction increasingly requires automated systems that recognize rich, interacting psychological states (e.g., emotions, personality traits, ambivalence/hesitancy) from multimodal, in-the-wild data. Despite mounting studies of cross-task correlations (Li et al., 2022; Wang et al., 2023), such as personality-guided Emotion Recognition (ER) (Wen et al., 2024) or emotion-informed personality modeling (Bao et al., 2025), the field predominantly deploys single-task, single-corpus architectures (Li et al., 2023; Kong et al., 2025). Recent advances in psychological states recognition (see detailed related work in Appendix A.1) have largely progressed in isolation: State-of-the-Art (SOTA) methods for Personality Traits Recognition (PTR) benefit from attention-based modeling of Big Five traits (Agrawal et al., 2023; Masumura et al., 2025). Ambivalence/Hesitancy Recognition (AHR) relies on temporal modeling via Temporal Convolutional Networks (TCNs) and Long Short-Term Memorys (LSTMs) (Kollias et al., 2025; Hallmen et al., 2025). Moreover, ER increasingly leverages Transformer and Mamba architectures for multimodal fusion (Goncalves et al., 2023; Zhang et al., 2025a). It is known that the correlation between various tasks of affective computing can enhance the model's performance. For instance, personality traits such as Neuroticism demonstrably modulate emotional reactivity to negative stimuli (Mohammadi & Vuilleumier, 2022). At the same time, ambivalence serves as a critical indicator of internal conflict,

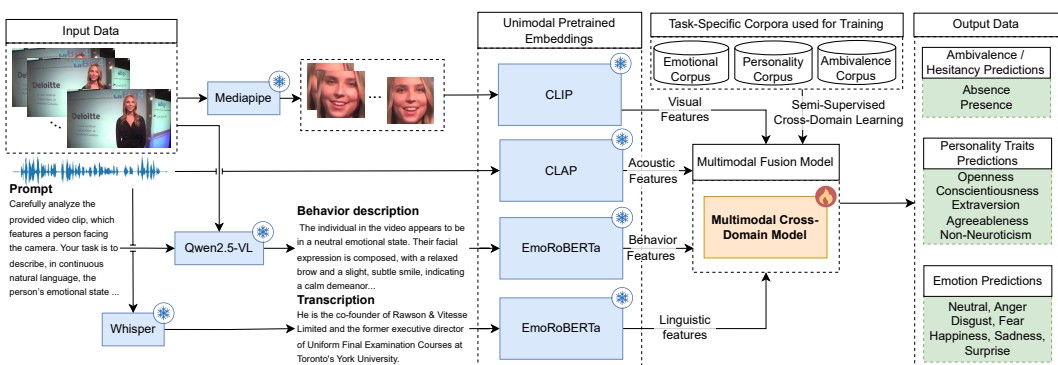

Figure 1: Pipeline of the proposed SCD-MMPSR method.

revealing whether an expressed emotion is genuine or socially masked, or whether self-reported personality aligns with behavioral cues (Hallmen et al., 2025). However, although multitask methods are emerging, particularly in emotion-sentiment or personality-emotion settings (Wen et al., 2024; Bao et al., 2025), they remain constrained to single corpora or homogeneous annotations. Meanwhile, Semi-Supervised Learning (SSL) has gained traction as a solution to annotation scarcity, with successful applications in unimodal ER, PTR (Hosseini & Caragea, 2023; Zhu et al., 2024) and multimodal methods (Fan et al., 2024; Lian et al., 2024).

Thus, nearly all effective methods remain task- and corpus-specific. These methods are trained using single-task corpora with narrow annotation scopes, inconsistent recording conditions, and different labeling protocols. This fragmentation imposes severe problems: (1) computational inefficiency, as deploying separate models for task-specific recognition multiplies inference overhead; and (2) poor generalization, as models trained on narrow, task-specific corpora suffer from domain overfitting and fail to transfer to unseen contexts. However, the lack of a general multitask solution stems from the fact that manual annotation at large-scale data is prohibitively expensive and often infeasible due to the complexity of the tasks and inter-annotator disagreement (Kollias et al., 2025; Sun et al., 2025; Mendelman & Talmon, 2025). Consequently, joint modeling of these states has remained largely unexplored, and the field lacks a practical, principled method to exploit many single-task, heterogeneous corpora jointly to learn shared multimodal representations that transfer across various affective behavior understanding tasks.

To fill this gap, we introduce SCD-MMPSR (**S**emi-supervised **C**ross-**D**omain **M**ultitask **M**ultimodal **P**sychological **S**tates **R**ecognition), a unified framework (Figure 1) that enables joint training across heterogeneous, single-task corpora without requiring joint annotations. We rigorously evaluate SCD-MMPSR on three benchmark corpora (CMU Multimodal Opinion Sentiment and Emotion Intensity (MOSEI) (Bagher Zadeh et al., 2018) annotated for ER, ChaLearn First Impressions v2 (FIv2) (Escalante et al., 2020) annotated for PTR, and Behavioural Ambivalence/Hesitancy (BAH) (González-González et al., 2025)annotated for AHR) under standard protocols, and further test its zero-shot generalization on the unseen Multimodal EmotionLines Dataset (MELD) corpus (Poria et al., 2019). The results demonstrate high generalization due to multitask SSL, validating the framework's capacity for cross-domain and cross-task transfer learning.

The main contributions of the article are as follows:

- SCD-MMPSR, an open-source semi-supervised cross-domain learning framework that jointly models ER, PTR, and AHR from heterogeneous, single-task corpora by using a GradNorm-based adaptive task weighting in multitask SSL.

- A Multimodal Cross-Domain Model (MCDM) with novel layers to learn cross-modal and cross-task interaction, called (1) Task-Specific Projectors for iterative feature-prediction refinement and (2) Guide Banks for structuring semantic task-specific embedding prototypes.

- Empirical evidence that our semi-supervised cross-domain learning improves multitask performance and generalization across various corpora, supported by ablations that isolate the benefits of the proposed modules.

## 2 PROPOSED METHOD

The proposed SCD-MMPSR (Figure 1) is a unified framework designed to predict ambivalence/hesitancy jointly, Personality Traits (PTs), and emotions from video sequences. Each video is divided into four modalities: video, audio, text, and behavior, which are then processed separately through specific pre-processing pipelines. Each modality is encoded using specialized pre-trained models to capture domain-specific features. These unimodal embeddings are then combined in a multimodal fusion architecture called MCDM. This model enhances cross-modal alignment and improves cross-domain generalization by utilizing task-specific corpora simultaneously. A key advantage of the proposed method is its unified architecture for multitask learning, which enables joint optimization across different tasks. Training is performed in a cross-domain setting, where each task uses its own corpus to create robust and transferable representations. A pseudo-labeling technique is used to leverage unlabeled data, which involves applying a confidence threshold to high-confidence predictions and integrating them into the training process in a semi-supervised manner. This enhances generalization without requiring additional annotation effort.

### 2.1 PRE-TRAINED EMBEDDINGS

In this study, we investigate the generalization of pre-trained encoders across different modalities and tasks using a unified multimodal framework. Instead of developing new encoders from scratch, we utilize existing models that have demonstrated effectiveness in affective and behavioral analysis (Appendix A.1). Systematically replacing one encoder at a time, while keeping the rest fixed, enables us to assess the contribution of each component to cross-modal and cross-task performance in a controlled manner. The evaluation is conducted in a multitask setting to assess the robustness of the encoders beyond the limitations of single-task scenarios.

We examine a range of encoders across four modalities: audio, video, and text/behavior. For audio, we use CLAP (Wu et al., 2023), Whisper (Radford et al., 2023), AST (Gong et al., 2021), and Wav2Vec2 (Baevski et al., 2020) models, including emotion-fine-tuned versions, as well as EmoEx-HuBERT (Amiriparian et al., 2024) and EmoWav2Vec2 (Wagner et al., 2023). Text and behavior encodings use both general-purpose models such as Jina (V3 / V4) (Sturua et al., 2024; Günther et al., 2025), BGE (Xiao et al., 2024a), CLAP (Wu et al., 2023), CLIP (Radford et al., 2021), and RoBERTa (Liu et al., 2019), as well as its modifications such as XLM RoBERTa (Conneau et al., 2019) and the affective models, such as EmoDistilRoBERTa (Sanh et al., 2019) and EmoRoBERTa[1]. For video, we study Dino v2 (Oquab et al., 2024), CLIP (Radford et al., 2021) and ViT (Wu et al., 2020), ResNet-50 (He et al., 2016), and emotion-specific models such as EmotiEffLib (Savchenko, 2023), EmoAffectNet (Ryumina et al., 2022), and two EmoViT models.

Pre-processing is applied to all modalities before encoding. For videos, the BlazeFace model (Bazarevsky et al., 2019) is used to detect face regions for accurate long-range tracking. This is followed by alignment and background removal using the FaceMesh model (Kartynnik et al., 2019). Both models are available in the MediaPipe library (Lugaresi et al., 2019), and their combined use allows eliminating each other's limitations. Audio signals are encoded directly with the selected pre-trained models, without any additional normalization. Text transcription is extracted using the Whisper Turbo model (Radford et al., 2023) and is fed to the encoder.

### 2.2 LLM-BASED BEHAVIOR DESCRIPTION

In recent research, the use of Large Visual Language Models (VLLMs) to describe human behavior in videos has been shown to enhance affect recognition performance (Zhang et al., 2024a; Lu et al., 2025). In our work, we use the Qwen2.5-VL-3b model (Bai et al., 2025) to generate video behavior descriptions, as it provides robust fine-grained visual comprehension, long-term video reasoning, and adaptive resolution. The prompt design for our experiments is based on the following idea. Instead of listing specific categories, the prompt encourages the model to generate continuous natural language narratives of observed behavior. It focuses on non-verbal cues, such as eye gaze, body posture, and microexpressions, and it avoids making assumptions about the context that cannot be verified. This narrative-based prompting ensures consistency across emotional, personality-related,

---

[1] https://huggingface.co/michellejieli/emotion_text_classifier

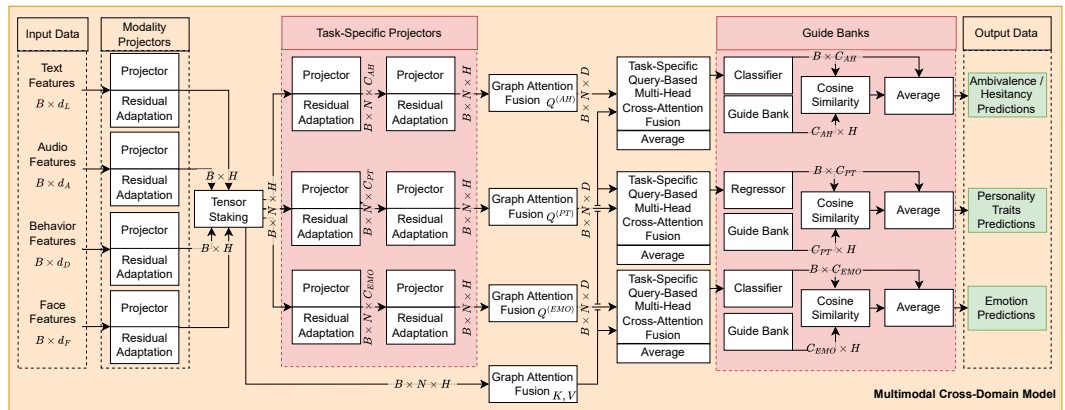

Figure 2: MCDM Architecture.

and ambiguous states, aligning with established psychological theories while leveraging the generative capabilities of VLLMs. The proposed prompt is presented in Appendix A.2. To confirm the effectiveness of the proposed prompt, we compare it with prompts designed explicitly for ER (Cheng et al., 2024; Zhang et al., 2025b).

## 2.3 MULTIMODAL CROSS-DOMAIN MODEL ARCHITECTURE

Since no existing corpus is jointly annotated for all three target tasks, we use multiple corpora from different domains. These differ in their recording conditions, annotation protocols, and label distributions. Importantly, the informativeness of modalities varies significantly across these corpora. To address this, we have designed a unified architecture that dynamically allocates attention across modalities within each task domain, while enabling cross-modal feature refinement to capture complementary signals. The architecture of the proposed MCDM is shown in Figure 2. MCDM addresses multimodal fusion across heterogeneous inputs by combining Modality- and Task-Specific Projectors, graph attention, task-specific query-based cross-attention fusions, and task-guided embedding banks. Each model component has its own purpose. The model therefore maps unimodal features $\{\boldsymbol{X}^{(m)}\}_{m\in\mathbb{M}}$ to task-specific predictions $\{\hat{\boldsymbol{y}}^{(t)}\}_{t\in\mathbb{T}}$ via a unified architecture. Let $\mathbb{M}$ denote the set of active modalities, and $\mathbb{T} = \{\text{EMO}, \text{PT}, \text{AH}\}$ the set of recognition tasks. For each modality $m \in \mathbb{M} = \{F(\text{video}), D(\text{behavior}), A(\text{audio}), L(\text{text})\}$, the input is a tensor $\boldsymbol{X}^{(m)} \in \mathbb{R}^{B \times d_m}$, where $B$ is the batch size and $d_m$ is the input feature dimension. These input tensors are statistical functionals (mean and standard deviation) calculated from contextual embeddings extracted using unimodal encoders. Each modality is then mapped into a shared hidden dimension space $H$ via a modality projector. The modality projector ensures that heterogeneous unimodal embeddings are mapped into a unified latent space while retaining modality-specific inductive bias. It is calculated as:

$$\boldsymbol{z}^{(m)} = \phi_m(\boldsymbol{X}^{(m)}) \in \mathbb{R}^{B \times H}, \tag{1}$$

where $\phi_m(\cdot)$ consists of a Fully Connected Layer (FCL), a Rectified Linear Unit (ReLU) activation function, a dropout layer, and a residual adaptation, which is calculated using the formula:

$$\tilde{\boldsymbol{z}}^{(m)} = \text{LayerNorm}(\boldsymbol{z}^{(m)} + \text{Adapter}(\boldsymbol{z}^{(m)})) \tag{2}$$

where Adapter$(\cdot)$ consists of a downsampling FCL (with weight tensor $\boldsymbol{W}_{down} \in \mathbb{R}^{H \times H/2}$), ReLU, a dropout layer, and an upsampling FCL ($\boldsymbol{W}_{up} \in \mathbb{R}^{H/2 \times H}$).

Concatenating across modalities yields the fused tensor:

$$\boldsymbol{Z} = \text{stack}(\tilde{\boldsymbol{z}}^{(m)})_{m\in\mathbb{M}} \in \mathbb{R}^{B \times N \times H}, \tag{3}$$

where $N = |\mathbb{M}|$ denotes the number of active modalities.

The graph attention fusion, GAF$(\cdot)$, is then applied to both modality features and Task-Specific Projectors. The modality features are processed by the shared GAF$(\cdot)$ to yield a unified key-value

representation that contains general representations for all task-specific domains. At the same time, the Task-Specific Projectors outputs are then refined through three GAF$(\cdot)$ layers to perform intra-modal and intra-task message passing in order to generate contextualized and task-aware queries. Given an adjacency $\boldsymbol{A} \in \{0,1\}^{B \times N \times N}$, the graph attention operator updates node embeddings as follows:

$$\text{GAF}(\boldsymbol{Z}, \boldsymbol{A})_{b,i,:} = \sum_{j=1}^{N} \alpha_{b,ij} \boldsymbol{W} \boldsymbol{Z}_{b,j,:} \in \mathbb{R}^{B \times N \times H}, \tag{4}$$

with attention coefficients:

$$\alpha_{b,ij} = \frac{\exp\big(\text{LeakyReLU}\big(\boldsymbol{a}^\top [\boldsymbol{W} \boldsymbol{Z}_{b,i,:} \,\|\, \boldsymbol{W} \boldsymbol{Z}_{b,j,:}]\big)\big) \, \mathbf{1}_{\{\boldsymbol{A}_{b,ij}>0\}}}{\sum_{j'} \exp\big(\text{LeakyReLU}\big(\boldsymbol{a}^\top [\boldsymbol{W} \boldsymbol{Z}_{b,i,:} \,\|\, \boldsymbol{W} \boldsymbol{Z}_{b,j',:}]\big)\big) \, \mathbf{1}_{\{\boldsymbol{A}_{b,ij'}>0\}}}, \tag{5}$$

where $\boldsymbol{a} \in \mathbb{R}^{2H}$ is a learnable parameter and $\|\cdot\|$ denotes concatenation; $b \in B$ is a batch index; $\boldsymbol{W} \in \mathbb{R}^{H \times H}$ is a weight tensor. When graph connections are disabled, the identity operator is used instead.

**Task-Specific Projectors functional**. For each task $t \in \mathbb{T}$, per-modality predictions are obtained as follows:

$$\boldsymbol{L}^{(t)} = \phi_t(\boldsymbol{Z}) \in \mathbb{R}^{B \times N \times C_t}, \tag{6}$$

with $C_t$ task-specific output dimension. These predictions are projected back to the hidden space:

$$\boldsymbol{P}^{(t)} = \phi_t(\boldsymbol{L}^{(t)}) \in \mathbb{R}^{B \times N \times H}, \tag{7}$$

and refined with a second graph operator:

$$\boldsymbol{C}_{\text{preds}}^{(t)} = \text{GAF}^{(t)}(\boldsymbol{P}^{(t)}, \boldsymbol{A}^{(t)}), \tag{8}$$

where $\boldsymbol{C}_{\text{preds}}^{(t)}$ are the contextualized prediction embeddings; both tensors, $\boldsymbol{L}^{(t)}$ and $\boldsymbol{P}^{(t)}$, pass through a task-specific projector, $\phi_t$, similar to a multimodal projector, $\phi_m$. The Task-Specific Projectors map shared multimodal embeddings into task-conditioned logits and re-embed them into the hidden space. This allows predictions to be refined through graph message passing and aligned with modality features via cross-attention.

Contextualized modality features are obtained analogously as $\boldsymbol{C}_{\text{mods}} = \text{GAF}(\boldsymbol{Z}, \boldsymbol{A}^{\text{feat}})$. Task-specific query-based cross-attention fusion, AF$(\cdot)$, integrates the two: with $\boldsymbol{C}_{\text{preds}}^{(t)}$ as queries $(Q^{(t)})$ and $\boldsymbol{C}_{\text{mods}}$ as keys / values $(K, V)$, we compute:

$$\boldsymbol{T}^{(t)} = \text{AF}\big(\boldsymbol{C}_{\text{preds}}^{(t)}, \boldsymbol{C}_{\text{mods}}, \boldsymbol{C}_{\text{mods}}\big), \tag{9}$$

AF$(\cdot)$ aligns contextualized prediction embeddings with modality features to reinforce task-specific feature representations. These task-specific representations are averaged across modalities:

$$\boldsymbol{r}^{(t)} = \frac{1}{N} \sum_{i=1}^{N} \boldsymbol{T}_{:,i,:}^{(t)} \in \mathbb{R}^{B \times H}. \tag{10}$$

The final logs are produced through the task heads $h_t(\cdot)$:

$$\hat{\boldsymbol{y}}_{\text{head}}^{(t)} = h_t(\boldsymbol{r}^{(t)}) \in \mathbb{R}^{B \times C_t}. \tag{11}$$

**Guide Bank functional**. In the Guide Banks, each task $t$ maintains embeddings $\boldsymbol{G}^{(t)} \in \mathbb{R}^{C_t \times H}$. These embeddings are learnable class prototypes, randomly initialized and dynamically updated during training. The input representation $\boldsymbol{r}_{b,:}^{(t)} \in \mathbb{R}^H$ is the output of the task-specific cross-attention module for batch sample $b$, i.e., the refined multimodal feature vector before the final prediction head. Cosine similarity between this representation and each prototype yields a semantic alignment score:

$$\text{sim}_{b,c} = \cos\big(\boldsymbol{r}_{b,:}^{(t)}, \boldsymbol{G}_{c,:}^{(t)}\big). \tag{12}$$

where $c$ indexes the class for task $t$. Each similarity score $\text{sim}_{b,c}$ reflects the degree to which sample $b$ conforms to the semantic prototype of class $c$. The Guide Banks introduce a structured semantic prior by anchoring predictions to task-specific embedding prototypes. This stabilizes learning

and improves generalization. The final prediction is a combination of the head outputs and guide similarities:

$$\hat{\boldsymbol{y}}^{(t)} = \begin{cases} \frac{1}{2}\left(\hat{\boldsymbol{y}}_{\text{head}}^{(t)} + \text{sim}\right), & t \neq \text{PT}, \\ \frac{1}{2}\left(\sigma(\hat{\boldsymbol{y}}_{\text{head}}^{(t)}) + \sigma(\text{sim})\right), & t = \text{PT}, \end{cases} \tag{13}$$

where $\sigma(\cdot)$ is the logistic sigmoid function, which is only applicable to PTR, as the values of the PTs scores range from 0 to 1. In our work, we compare the performance of various Graph Neural Network (GNN), including vanilla GNN (Veličković et al., 2018), Non-Convolutional GNN (NCGNN) (Wang & Cho, 2024), Unitary Convolutions GNN (UCGNN) (Kiani et al., 2024), Edge Directions GNN (EDGNN) (Pahng & Hormoz, 2025), Hyperbolic GNN (HGNN) (Yue et al., 2025) and attention mechanisms, including Multi-Head Attention (MHA) (Vaswani et al., 2017), Bidirectional Cross Attention (BiCA) (Hiller et al., 2024), Cross-attention Message-Passing Transformer (CrossMPT) (Park et al., 2025), Multi-Token Attention (MTA) (Golovneva et al., 2025), Forgetting Attention (FA) (Lin et al., 2025) to determine the optimal model configuration.

In this paper, we explicitly differentiate between two complementary components: SCD-MMPSR and MCDM. SCD-MMPSR denotes the full framework (including semi-supervised cross-domain learning protocol and data pre-processing), while MCDM refers to its central multimodal fusion model.

## 2.4 SEMI-SUPERVISED CROSS-DOMAIN LEARNING

We use three task-specific corpora, each of which is annotated exclusively for one task: ER, PTR, or AHR. Each corpus provides labels only for its own task, while the remaining labels are set to None. During training, a batch is constructed by randomly sampling from all corpora. Let $n_1, n_2, n_3$ be the randomly selected samples drawn from the three corpora, with batch size $B = n_1 + n_2 + n_3$.

We use a hybrid loss with adaptive task weighting, based on an extended GradNorm method (Chen et al., 2018). For each task, we define a supervised loss ($\mathcal{L}_\text{s}$) applied only to labeled samples, while unlabeled samples are masked out:

$$\mathcal{L}_\text{s} = w_\text{EMO}^\text{s}\,\mathcal{L}_\text{EMO}^\text{s} + w_\text{PT}^\text{s}\,\mathcal{L}_\text{PT}^\text{s} + w_\text{AH}^\text{s}\,\mathcal{L}_\text{AH}^\text{s}, \tag{14}$$

where $\mathcal{L}_\text{EMO}^\text{s}$ is Cross-Entropy (CE) loss for ER, $\mathcal{L}_\text{PT}^\text{s}$ is Mean Absolute Error (MAE) loss for PTR, and $\mathcal{L}_\text{AH}^\text{s}$ is CE loss for AHR. The weights $\{w_t^\text{s}\}_{t\in\mathbb{T}}$ are not fixed hyperparameters, but are dynamically optimized during training to balance gradient magnitudes across tasks.

To exploit unlabeled samples, we use pseudo-labeling with confidence thresholds. Pseudo-labels are generated in the same forward pass as the supervised loss, without the need for a separate teacher model or exponential moving average updates. We use a pseudo-label scheme because our dual-branch GradNorm mechanism adaptively balances supervised and semi-supervised losses, enabling stable SSL of the proposed model. For ER and AHR, pseudo-labels are assigned from the softmax probabilities if the maximum confidence exceeds $\tau_\text{EMO/AH}$. For PTR, logits are binarized at $0.5$ (as a threshold value for the PTs polarity) and accepted as pseudo-labels if they fall outside the uncertainty margin, i.e., if they are above $\tau_\text{PT}$ or below $1 - \tau_\text{PT}$. The semi-supervised loss ($\mathcal{L}_\text{ss}$) is then computed as:

$$\mathcal{L}_\text{ss} = w_\text{EMO}^\text{ss}\,\mathcal{L}_\text{EMO}^\text{ss} + w_\text{PT}^\text{ss}\,\mathcal{L}_\text{PT}^\text{ss} + w_\text{AH}^\text{ss}\,\mathcal{L}_\text{AH}^\text{ss}, \tag{15}$$

where $\mathcal{L}_\text{EMO}^\text{ss}$ is CE loss for ER, $\mathcal{L}_\text{PT}^\text{ss}$ is Binary CE (BCE) loss for PTR, and $\mathcal{L}_\text{AH}^\text{ss}$ is CE loss for AHR. The total hybrid loss combines both components $\mathcal{L} = \mathcal{L}_\text{s} + \mathcal{L}_\text{ss}$. Task weights $w_t^\text{s}$ and $w_t^\text{ss}$ are updated online through two independent GradNorm branches, which minimize auxiliary balancing losses:

$$\mathcal{L}_\text{GradNorm}^\text{s} = \sum_{t\in\mathbb{T}}\left|G_t^\text{s} - \overline{G}^\text{s}\cdot(r_t^\text{s})^{\alpha^\text{s}}\right|, \quad \mathcal{L}_\text{GradNorm}^\text{ss} = \sum_{t\in\mathbb{T}}\left|G_t^\text{ss} - \overline{G}^\text{ss}\cdot(r_t^\text{ss})^{\alpha^\text{ss}}\right|, \tag{16}$$

where for each task $t$ and branch (supervised or semi-supervised). $G_t = \|\nabla_{\theta_\text{shared}}(w_t\cdot\mathcal{L}_t)\|_2$ is the $\ell_2$-norm of the gradient of the weighted task loss with respect to shared model parameters $\theta_\text{shared}$. $\overline{G} = \frac{1}{|\mathbb{T}|}\sum_{j\in\mathbb{T}}G_j$ is the mean gradient norm across all tasks $\mathbb{T}$ in the current branch (supervised or SSL). $r_t = \frac{\mathcal{L}_t/\mathcal{L}_t^{(0)}}{\frac{1}{|\mathbb{T}|}\sum_{j\in\mathbb{T}}\mathcal{L}_j/\mathcal{L}_j^{(0)}}$ is the relative inverse training rate, comparing the normalized loss of task $t$ to the branch-wise average. Here, $\mathcal{L}_t$ denotes the raw, unweighted loss for the task $t$ (with

$\mathcal{L}_t$ being supervised ($\mathcal{L}_t^{\text{s}}$) or semi-supervised ($\mathcal{L}_t^{\text{ss}}$), depending on the branch), and $\mathcal{L}_t^{(0)}$ is its value recorded at the first training step where it became finite and valid – serving as a per-task initialization baseline. $\alpha^{\text{s}}$ and $\alpha^{\text{ss}}$ control the aggressiveness of balancing, with larger $\alpha$ penalizing faster-learning tasks more strongly. The task weights are then updated via gradient descent on $\mathcal{L}_{\text{GradNorm}}$ with task-type-specific learning rates:

$$w_t^{\text{s}} \leftarrow \max\left(w_{\text{floor}},\ w_t^{\text{s}} - \eta_w^{\text{s}} \cdot \nabla_{w_t^{\text{s}}} \mathcal{L}_{\text{GradNorm}}^{\text{s}}\right), \tag{17}$$

$$w_t^{\text{ss}} \leftarrow \max\left(w_{\text{floor}},\ w_t^{\text{ss}} - \eta_w^{\text{ss}} \cdot \nabla_{w_t^{\text{ss}}} \mathcal{L}_{\text{GradNorm}}^{\text{ss}}\right), \tag{18}$$

where separate learning rates ($\eta_w^{\text{s}}, \eta_w^{\text{ss}}$) with $w_{\text{floor}}$ preventing any task from being deactivated. After each update, weights are renormalized to budgets to enforce interpretable task prioritization:

$$w_t^{\text{s}} \leftarrow S^{\text{s}} \cdot \frac{w_t^{\text{s}}}{\sum_{j \in \mathbb{T}} w_j^{\text{s}}}, \quad w_t^{\text{ss}} \leftarrow S^{\text{ss}} \cdot \frac{w_t^{\text{ss}}}{\sum_{j \in \mathbb{T}} w_j^{\text{ss}}}, \tag{19}$$

with budgets $S^{\text{s}} = 3.0$ and $S^{\text{ss}} = 3.0 \times \lambda$, reflecting higher initial priority for supervised signals.

The adaptive change of task contribution coefficients serves as an implicit form of gradient-aware regularization. By aligning the task-specific gradients with their relative progress during training, GradNorm promotes balanced optimization and prevents the dominance of noisy or overfitting tasks, which is critical in semi-supervised cross-domain learning. Our dual-branch extension improves upon standard GradNorm in three key ways: (1) it decouples supervised and SSL optimization to account for differing noise levels; (2) it delays weight initialization until valid losses appear, to handle missing labels; and (3) it enforces explicit budget constraints for interpretable prioritization. The thresholds $\{\tau_t\}_{t \in \mathbb{T}}$ and other coefficients ($\alpha^{\text{s}}, \alpha^{\text{ss}}, \eta_w^{\text{s}}, \eta_w^{\text{ss}}, w_{\text{floor}}$, and $\lambda$) remain hyperparameters tuned on validation data, while the task contribution coefficients ($w_t^{\text{s}}, w_t^{\text{ss}}$) are now fully adaptive, eliminating manual tuning and improving robustness to dynamic label imbalance.

This hybrid loss function SSL across single-task corpora by combining supervised objectives with pseudo-labeled consistency. This alleviates task-wise label sparsity and improves cross-task generalization. The function is further stabilized by gradient-aware adaptive weighting.

## 3 EXPERIMENTS

### 3.1 CORPORA

In Appendix A.3, we provide a summary of existing corpora and identify corpora applicable to our study. We use three task-specific corpora, each of which is annotated for a single objective. For ER, we use the MOSEI corpus (Bagher Zadeh et al., 2018), the largest multimodal corpus for affect analysis. This contains over 23,500 YouTube videos at the utterance-level from more than 1,000 speakers, which have been annotated for six basic emotions (Anger, Disgust, Fear, Happiness, Sadness, Surprise). Each video may have multiple labels, with no labels indicating a neutral state. For PTR, we use the FIv2 corpus (Escalante et al., 2020), which comprises 10,000 short vlogs (15 seconds each) from approximately 3,000 speakers. Each clip is annotated for the Big Five PTs (Openness, Conscientiousness, Extraversion, Agreeableness, Non-Neuroticism), with continuous scores between 0 and 1 obtained via pairwise comparisons. Finally, for AHR, we adopt the recently introduced BAH corpus (González-González et al., 2025), comprising 1,118 video recordings from 224 participants across nine Canadian provinces. The corpus is annotated for two categories: Absence or Presence of ambivalence/hesitancy. The corpora are split into Train, Development, and Test subsets. In Appendix A.4, we present the distribution of classes in subsets. Each corpus provides supervision only for its designated task, creating a heterogeneous setup in which cross-task generalization is enabled by SSL with pseudo-labels. To evaluate the cross-dataset generalizability of SCD-MMPSR to unseen data, we utilize MELD (Poria et al., 2019), a dataset of video recordings from the TV series "Friends". The corpus has been annotated for six basic emotions and a neutral state. We only use the fixed Test subset from this corpus. Performing model generalization assessment is difficult for other tasks due to the lack of corpora with a similar annotation protocol.

### 3.2 EXPERIMENTAL SETUP

We design a multi-stage experimental protocol to systematically assess the proposed framework. First, we identify the most effective unimodal encoders within a unified multimodal system (see

Table 1: Experimental results of ablation studies. MCDM means the proposed Multimodal Cross-Domain Model. MCDM-1 based on vanilla GNN and MHA. MCDM-2 based on Unitary Convolutions GNN (UCGNN) and MHA. MCDM-3 based on Unitary Convolutions GNN (UCGNN) and Multi-Token Attention (MTA). V, A, T, and B stand for Video, Audio, Text, and Behavior. Rank is calculated using Friedman's test (Demšar, 2006). Best and second-best results are highlighted

| Exp ID | Extractors | Model | MOSEI | | FIv2 | | BAH | | MELD | | Rank |
|---|---|---|---|---|---|---|---|---|---|---|---|
| | | | mMF1 | mWACC | mACC | mCCC | MF1 | UAR | MF1 | WF1 | |
| 1 | V+CLIP, A+CLAP, T+CLAP, B+CLIP | MCDM-1 | 61.50 | 61.87 | 91.46 | 66.10 | 65.66 | 65.36 | 30.91 | 38.56 | 13.13 |
| 2 | V+CLIP, A+CLAP, T+EmoRoBERTa, B+EmoRoBERTa | MCDM-1 | 63.40 | 64.00 | 91.44 | 66.68 | 69.29 | 69.07 | 33.66 | 40.04 | 8.38 |
| 3 | V+CLIP, A+CLAP, T+EmoRoBERTa, B+EmoRoBERTa | MCDM-2 | 63.35 | 63.99 | 91.67 | 69.38 | 69.14 | 69.12 | 34.06 | 42.53 | 6.75 |
| 4 | V+CLIP, A+CLAP, T+EmoRoBERTa, B+EmoRoBERTa | MCDM-3 | 63.63 | 64.42 | 91.42 | 66.84 | 67.52 | 68.44 | 34.36 | 40.06 | 8.13 |
| 5 | Exp-3 and best hyperparameters (Appendix A.7) | MCDM-2 | 63.06 | 63.62 | 91.77 | 69.51 | 70.38 | 70.28 | 36.26 | 42.51 | 5.25 |
| 6 | Exp-5 and best SSL parameters (Appendix A.8) | MCDM-2 | 63.14 | 63.61 | 91.93 | 71.98 | 71.70 | 71.42 | 35.04 | 44.08 | 3.25 |
| 7 | Exp-6 w/o Task-Specific Projectors | MCDM-2 | 62.05 | 63.47 | 91.81 | 71.11 | 70.20 | 70.60 | 34.17 | 42.45 | 6.38 |
| 8 | Exp-6 w/o Graph Layers | MCDM-2 | 61.92 | 62.24 | 91.87 | 73.03 | 71.12 | 71.17 | 32.99 | 39.48 | 7.38 |
| 9 | Exp-6 w/o Attention Layers | MCDM-2 | 60.02 | 61.71 | 91.51 | 69.30 | 37.88 | 50.00 | 23.22 | 33.65 | 13.88 |
| 10 | Exp-6 w/o Guide Bank Layers | MCDM-2 | 62.35 | 62.88 | 91.97 | 73.21 | 70.12 | 70.74 | 31.87 | 39.88 | 6.13 |
| 11 | Exp-6 w/o Video Modality | MCDM-2 | 61.91 | 62.69 | 90.32 | 57.87 | 67.73 | 68.01 | 20.64 | 19.13 | 13.75 |
| 12 | Exp-6 w/o Audio Modality | MCDM-2 | 62.02 | 62.74 | 91.61 | 70.44 | 69.08 | 69.41 | 34.54 | 45.41 | 7.63 |
| 13 | Exp-6 w/o Text Modality | MCDM-2 | 57.62 | 58.69 | 91.95 | 73.35 | 62.47 | 63.43 | 15.11 | 23.57 | 12.00 |
| 14 | Exp-6 w/o Behavior Modality | MCDM-2 | 61.95 | 62.08 | 91.76 | 72.62 | 71.49 | 71.62 | 27.97 | 32.52 | 8.50 |
| 15 | Exp-6 w/o ER task | MCDM-2 | – | – | 91.76 | 71.88 | 69.14 | 69.53 | – | – | 9.25 |
| 16 | Exp-6 w/o PTR task | MCDM-2 | 62.92 | 63.61 | – | – | 70.46 | 69.95 | 29.40 | 35.95 | 7.83 |
| 17 | Exp-6 w/o AHR task | MCDM-2 | 62.10 | 62.46 | 91.88 | 73.05 | – | – | 33.78 | 41.49 | 6.67 |

Appendix A.5). As base extractors, we employ CLIP (Radford et al., 2021) (for video and scene descriptions) and CLAP (Wu et al., 2023) (for audio and transcripts), preserving semantic alignment across modalities as demonstrated in (Gan et al., 2023). We evaluate different model configurations with fixed encoders by replacing the graph layers and attention mechanisms (see Appendix A.6). The baseline model (MCDM-1) adopts the vanilla GNN (Veličković et al., 2018) and MHA (Vaswani et al., 2017). For the video, we compare different numbers of frames, while for behavior, we compare our prompt with two alternatives (Cheng et al., 2024; Zhang et al., 2025b) (see Appendix A.2).

Second, we construct two enhanced model configurations: MCDM-2 with a modified best-performing UCGNN (Kiani et al., 2024) and MCDM-3 with a modified best-performing UCGNN and MTA (Golovneva et al., 2025). At this stage, we tune model-level hyperparameters (learning rate, optimizer, dropout, hidden dimensions, output feature size, and number of attention heads) alongside SSL parameters (loss coefficients and pseudo-label thresholds). This stage determines the optimal architecture (see Appendix A.7) and SSL configuration (see Appendix A.8).

Third, we conduct ablation studies by selectively disabling model components, modalities, and tasks (see Table 1). To compare with SOTA methods, we also run single-task settings with and without SSL, varying the probability of incorporating unlabeled data (see Table 2). This stage establishes the contribution of each component of a model and the advantage of our framework over SOTA results. Finally, we conduct an inter-task correlation study and an error analysis to assess the effectiveness of joint multitask learning under semi-supervised conditions (see Appendix A.9).

We applied several performance measures to evaluate SCD-MMPSR. mean Accuracy (mACC) (Escalante et al., 2020), and mean Concordance Correlation Coefficient (mCCC) (Lin, 1989) are used for PTR on FIv2 as a regression task. mean Weighted Accuracy (mWACC) (Bagher Zadeh et al., 2018) and mean Marco F1-score (mMF1) (Bagher Zadeh et al., 2018) are applied for multi-label ER on the MOSEI corpus. Classical classification recognition measures (Marco F1-score (MF1), Weighted F1-score (WF1), and Unweighted Average Recall (UAR) are unitized for single-label ER and AHR on MELD and BAH, respectively.

## 3.3 RESULTS

The experimental results are presented in Table 1. Optimization of the encoders (Exp-2, details in Appendix A.5) improves performance compared to the baseline model (Exp-1). Extending the baseline model with the UCGNN (Kiani et al., 2024) (Exp-3) improves performance. However, modifying the model with a MTA (Golovneva et al., 2025) (Exp-4) leads to decreased performance, indicating sensitivity to the choice of attention scheme. Overall, the performance improvement of Exp-3 is mainly due to the PTR. Further optimization of the model hyperparameters (details in

Table 2: Comparison with single-task SOTA methods. The confidence intervals of SCD-MMPSR are calculated using the bootstrap resampling method (Tibshirani & Efron, 1993)

| Method | Modality | Learning type | Learning domain | Performance measure | |
|--------|----------|---------------|-----------------|-----|-----|
| MOSEI | | | | mWACC | mMF1 |
| Zhang et al. (2022) | Video, Audio, Text | Supervised | Single-domain | 51.2 | – |
| Peng et al. (2024) | Video, Audio, Text | Supervised | Single-domain | 66.4 | – |
| Ryumina et al. (2025) | Video, Audio, Text | Supervised | Single-domain | 69.3 | – |
| SCD-MMPSR w/o SSL and multitask | Video, Audio, Text, Behavior | Supervised | Single-domain | 63.6 [62.9, 64.3] | 63.3 [62.7, 64.0] |
| SCD-MMPSR w/o multitask | Video, Audio, Text, Behavior | Semi-supervised | Cross-domain | 68.9 [68.2, 69.6] | 69.3 [68.5, 70.0] |
| MELD (testing only) | | | | WF1 | MF1 |
| SCD-MMPSR w/o SSL and multitask | Video, Audio, Text, Behavior | Supervised | Single-domain | 27.0 [25.1, 28.9] | 22.8 [21.1, 24.7] |
| SCD-MMPSR w/o multitask | Video, Audio, Text, Behavior | Semi-supervised | Cross-domain | 30.4 [28.7, 32.4] | 27.5 [25.6, 29.9] |
| FIv2 | | | | mACC | mCCC |
| Zhao et al. (2023) | Video, Audio | Supervised | Single-domain | 91.7 | – |
| Wang et al. (2025) | Video, Audio, Text | Supervised | Single-domain | 92.1 | – |
| Gan et al. (2023) | Video, Text | Supervised | Single-domain | 92.6 | – |
| SCD-MMPSR w/o SSL and multitask | Video, Audio, Text, Behavior | Supervised | Single-domain | 91.8 [91.7, 92.0] | 74.0 [72.6, 75.2] |
| SCD-MMPSR w/o multitask | Video, Audio, Text, Behavior | Semi-supervised | Cross-domain | 92.6 [92.5, 92.8] | 77.2 [75.8, 78.5] |
| BAH | | | | WF1 | MF1 |
| Kollias et al. (2025) | Video, Audio, Text, Gesture | Supervised | Single-domain | 70.0 | – |
| Hallmen et al. (2025) | Video, Audio, Text | Supervised | Single-domain | 70.2 | – |
| Savchenko & Savchenko (2025) | Video, Audio, Text | Supervised | Single-domain | 71.0 | – |
| SCD-MMPSR w/o SSL and multitask | Video, Audio, Text, Behavior | Supervised | Single-domain | 72.9 [68.5, 77.2] | 71.5 [66.6, 76.0] |
| SCD-MMPSR w/o multitask | Video, Audio, Text, Behavior | Semi-supervised | Cross-domain | 73.2 [68.9, 77.8] | 72.1 [67.6, 76.4] |

Appendix A.7) and SSL parameters (details in Appendix A.8) has a positive impact on performance. While this comes at a slight cost to the ER performance, it improves one on other tasks.

The component-level ablation study (Exp 7-10) reveals that the attention mechanism is the most crucial component, while graph attention plays a secondary role. The proposed layers, Task-Specific Projectors, and Guide Banks are also essential, as they help with effective task alignment and information sharing across modalities. The modality-level ablation study (Exp 11-14) emphasizes the importance of video and text modalities in recognizing psychological states, highlighting the significance of both verbal and non-verbal communication. The task-level ablation study (Exp 15-17) shows that confidence estimation benefits from the presence of AHR, while removing the task improves performance on other tasks. Overall, the ablation study shows that all proposed framework components significantly improve the model's performance. The results on MELD show high generalization ability, achieving MF1 = 35.04 and WFI = 44.08.

Table 2 compares the single-task versions of the SCD-MMPSR framework with SOTA methods. In supervised and single-domain settings, SCD-MMPSR tends to underperform compared to the SOTA methods. However, there is a significant improvement when the model is applied in SSL and cross-domain learning settings, leveraging unlabeled data from non-target corpora. Bootstrap confidence intervals confirm that improvements obtained by SCD-MMPSR over the SOTA are statistically significant, as its upper bounds are higher than the SOTA results. Although our model does not outperform SOTA performance in ER, our results show that using unlabeled data, including corpora annotated for other paralinguistic tasks, improves model performance. This improvement is achieved without task-specific fine-tuning of encoders or the need for additional annotation.

For MELD, although performance is improved under single-task SSL, it did not achieve the level of models trained jointly across all three tasks. The reduction in measure MF1 was 7.5% (27.5 vs. 35.0). The relative decrease in measure WF1 was 13.7% (30.4 vs. 44.1). These results indicate that single-task models are prone to overfitting and have limited generalization to unseen data. In contrast, our proposed framework significantly improves model generalization, resulting in robust performance on new data.

Speaking about the computational cost of SCD-MMPSR, the real-time factor for processing 1 sec of multimodal data using MediaPipe, Qwen2.5-VL-3b, Whisper, CLIP, CLAP, EmoRoBERTa, and MCDM is 1.11 sec on an NVIDIA A100 GPU. Of this, 0.69 sec is consumed by Qwen2.5-VL-3b, which limits inference of SCD-MMPSR to the CPU only. The parameter count of MCDM grows quadratically with the number of tasks; the full model occupies 38.2 MB. Thus, while our framework demonstrates strong cross-dataset generalizability to unseen data, its main limitation is its reliance on VLLMs. However, if there are resource constraints, we suggest omitting the behavior modality.

This may result in a decrease in model performance of approximately 2% (depending on the task, see Table 1), but it will also reduce inference time by approximately 1.5 times.

As an additional limitation of our framework, we introduce a few task-specific hyperparameters, such as confidence thresholds for pseudo-labeling and balancing coefficients for GradNorm, beyond standard deep learning settings (e.g., learning rate and batch size). These parameters can affect the stability and cross-task balance of our model, but adaptive mechanisms reduce the need for manual tuning. In future work, we plan to explore self-tuning or automated strategies to further enhance the reproducibility and robustness of our model.

## 4 CONCLUSION

This paper presented SCD-MMPSR, a compact semi-supervised framework for joint multimodal recognition of psychological states that bridges heterogeneous, single-task corpora. SCD-MMPSR combines pre-trained unimodal encoders with a graph-attention fusion backbone and three improvements: (1) Task-Specific Projectors for iterative feature-prediction refinement; (2) Guide Banks for structuring semantic task-specific embedding prototypes; and (3) the dual-branch GradNorm method to adaptive task weighting in multitask SSL. We evaluate our framework on three task-specific corpora (MOSEI, FIv2, and BAH) under standard train-dev-test protocols, and demonstrate its generalization capability on MELD in a zero-shot cross-domain setup. Results show that joint multitask training improves generalization over single-task baselines. This confirms that our framework enables effective cross-domain learning without requiring full annotation across tasks, leveraging pseudo-labels and SSL instead. In future work, we plan to scale the framework to additional tasks and integrate contrastive learning to enhance cross-task generalization by explicitly aligning task-invariant representations.

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

## A  APPENDIX

### A.1  RELATED WORK

#### A.1.1  STATE-OF-THE-ART PSYCHOLOGICAL STATES RECOGNITION METHOD

In this brief review, we consider methods for ER, PTR, and AHR. Emotions reflect transient reactions, while PTs reflects stable dispositions. Ambivalence reveals the uncertainty that may influence both states, providing critical insight into human intentions and decisions. These tasks enable the creation of more nuanced and context-sensitive human-machine interaction systems that cover only one specific task beyond classical affective recognition methods.

**Emotion Recognition Methods**. Multimodal Emotion Recognition (MER) is a crucial part of research related to analyzing human emotional state. Recent studies have noted that Deep Neural Networks (DNNs) provide robust results from integrating different modalities (Deng et al., 2024). Different types of Transformer architectures are used in multimodal feature extraction. For instance, Goncalves et al. (2023) presented an audio-visual framework that utilizes conformer layers instead of ordinary Transformers. Li et al. (2021) integrated a pre-trained BERT model (Devlin et al., 2019) with a K-Nearest Neighbors (KNN) classifier during fine-tuning. This method addresses the distribution shifts between the source domain and the target domain, enabling more accurate classification in cross-domain tasks. However, the study relies on minimizing the cross-entropy loss, which often leads to unstable fine-tuning and poor generalization. Hazarika et al. (2020) proposed a network, based on the Transformer architecture, in which features for each modality are projected to two distinct sub-spaces: modality-invariant and modality-specific. Tsai et al. (2019) applied the directional pairwise cross-modal attention mechanism, which attends to interactions between unaligned multimodal sequences across different timesteps. Liu et al. (2025) leveraged a multilevel method based on a spatio-temporal vision Transformer to extract facial and body features. Mamba is another deep learning architecture outperforming conventional Transformers (Gu & Dao, 2023). Experiments have proven that Mamba-based models capture inter-modal interactions through a cross-modal mechanism, achieving better modal representations (Zhang et al., 2025a). Xinyue et al. (2025) introduced LensLLM method which enables early performance prediction of Large Language Models (LLMs) by analyzing signals from the initial phases of fine-tuning.

Several widely known methods, including graph-based (Joshi et al., 2022; Li et al., 2023), and hybrid methods based on Convolutional Neural Network (CNN) and Recurrent Neural Network (RNN) (Gao et al., 2024; Xiao et al., 2024b), are used for ER. For instance, Joshi et al. (2022) proposed a contextualized GNN-based method aimed to capture information via both inner and outer context. Zadeh et al. (2018a) proposed LSTM-based neural architecture using a multi-view gated memory that stores a history of cross-view interactions and integrates information from different modalities at different timesteps. Hosseini et al. (2024) showed that the combination of the network of Convolutional Neural Network-Long Short-Term Memory (CNN-LSTM) and Bidirectional Long

Short-Term Memory (BiLSTM) achieves a high performance when learning the features of the fusion. Farhadipour et al. (2025) used CNN along with Transformer architecture for extracting visual features. In contrast, Boitel et al. (2025) leveraged advanced Deep Learning techniques combining Semi-CNN and 3D-CNN to enhance the robustness of data and comprehensively improve the performance of various modalities.

**Personality Traits Recognition Methods**. In addition to ER, recognizing persons' PTs has gained popularity over the past few years. PTR is often based on the scores of a psychological model named Big Five or the OCEAN model (McCrae, 2020). Deep learning algorithms such as CNN, LSTM, and the Transformer model are broadly applicable in PTR (Zhao et al., 2023). For pairwise and simultaneous comparison of Personality Traits Assessment (PTA) Ryumina et al. (2024) proposed the Gated Siamese Fusion Network (GSFN), which enables the fusion of both hand-crafted and deep features across text, audio, and video-face modalities. Kong et al. (2025) used a cross-attention mechanism to improve both the proposed model's robustness and the audiovisual modality's performance. In particular, Masumura et al. (2025) proposed SOTA Transformer-based methods that address two tasks: assessing people's PT scores along with questionnaire-based item-level scores. Agrawal et al. (2023) highlighted the significance of the Transformer architecture, presenting the Forced Attention Transformer for tackling tasks related to PTR.

**Ambivalence/Hesitancy Recognition Methods**. The AHR task was first introduced in the 8th Affective Behavior Analysis in-the-Wild (ABAW) competition (Kollias et al., 2025). To solve this task, the BAH corpus (González-González et al., 2025) was collected and annotated. The baseline method (Kollias et al., 2025) combined TCN (Bai et al., 2018) with acoustic, linguistic, and visual features and used a co-attention block to aggregate multimodal features and to create a single embedding for each frame. Hallmen et al. (2025) proposed a method that integrated text, audio, and visual modalities, modeling temporal dependencies in audio and vision with LSTMs and applying a convolution-like temporal windowing mechanism for frame-level prediction. All modalities were fused through a Multi-Layer Perceptron (MLP). Savchenko & Savchenko (2025) developed a multimodal method that emphasized efficient facial models, applied early fusion across modalities, and refined predictions with blending and temporal smoothing.

**Multitask Recognition Methods**. Several recent studies are devoted to SOTA multitask unimodal or multimodal methods, exploring various affective states. For instance, Markitantov et al. (2025) explored the multitask method based on Label Encoder Fusion Strategy for both ER and Sentiment Recognition (SR). However, it is important to note that only a limited number of studies focus on the conjunction between ER, PTR, or AHR. Several works have been devoted to studying emotional state via PTR (Hosseini et al., 2023; Wen et al., 2024). Wen et al. (2024) studied emotions based on PTR in dialogue systems and investigated the personality-affected mood transition afterward. PTR guided by emotional analysis has also been widely investigated (Yuanchao et al., 2023; Bao et al., 2025). Bao et al. (2025) was the first to employ contrastive learning to increase precision and predictability in multimodal PTR. Transfer learning using Transformer-based architecture is another effective way to study the correlation between personality and emotions (Yuanchao et al., 2023). Some recent research is focused on the correlation between ER and PTR within the scope of physiological signals (Hosseini et al., 2023; Pant et al., 2023). For instance, Hosseini et al. (2023) presented a SOTA method applied to ER based on the level of bioelectric activity of the brain. Seikavandi et al. (2025) proposed MuMTAffec for multimodal multitask ER and PTR on a limited corpus annotated for the two target tasks.

### A.1.2 State-of-the-Art Methods based on Semi-Supervised Learning

SSL has emerged as a crucial strategy for addressing the significant challenge of limited labeled data in machine learning (Mendelman & Talmon, 2025; Sun et al., 2025). This strategy leverages small amounts of labeled data and larger pools of unlabeled data to improve model performance, mitigating the high cost and difficulty associated with extensive manual annotation. Widely-known SSL methods, such as FixMatch and MixMatch, are based on the idea of consistency regularization and pseudo-labeling (Sohn et al., 2020; Melnychuk et al., 2020). MixMatch uses soft pseudo-labels through averaging and sharpening for both labeled and unlabeled data. FixMatch, on the other hand, utilizes one-hot confident pseudo-labels and employs both weak and strong data augmentations, thereby simplifying the training procedure. Several studies have investigated the Mean Teacher method for Object Detection and Instance Segmentation (Deng et al., 2021; Alayrac et al., 2020;

Cao et al., 2023). The method leverages a teacher-student framework where the teacher acts as Exponential Moving Average (EMA) of the student model, which generates pseudo-labels on unlabeled target-domain images. Recent research demonstrates the application of SSL in diverse areas of human behavior modeling, including ER (Hosseini & Caragea, 2023; Wu et al., 2025; Deng et al., 2025; Alameer et al., 2025), PTR (Zhu et al., 2024), and other psychological states recognition (Takahashi et al., 2024; Skat-Rørdam et al., 2024). These studies primarily focus on unimodal methods, including text (Hosseini & Caragea, 2023; Zhu et al., 2024), video (Takahashi et al., 2024; Deng et al., 2025), and physiological signals (such as Electroencephalogram (EEG)) (Tao et al., 2024; Martin-Melero et al., 2024; Alameer et al., 2025). Several works presented at the MER workshops (Lian et al., 2023; 2024) explicitly focus on multimodal SSL, highlighting its importance, relevance, and complexity (Fan et al., 2024). A common strategy in such studies involves combining corpora to create a larger, unified corpus for training models on a single, specific task, extending data domains, and enhancing model robustness (Zhang et al., 2024b; Skat-Rørdam et al., 2024). For instance, methods often integrate data augmentation (Zhu et al., 2024; Skat-Rørdam et al., 2024) or employ self-supervised and contrastive learning (Fan et al., 2024) within the SSL framework to enhance performance on unified tasks. However, while combining corpora for a single task is well-established, integrating corpora with distinct annotation tasks (e.g., emotion vs. PTs) and different domains (e.g., varying recording setups or participant demographics) within a semi-supervised cross-domain learning framework remains unknown.

Table 3 systematically compares the SOTA methods for recognition of different psychological states, including ER, PTR, and AHR. The analysis reveals the following trends. Linguistic features are predominantly extracted using Transformer-based encoders such as BERT or RoBERTa, reflecting their dominance in contextual language modeling. Acoustic representations rely on self-supervised models like Wav2Vec2 and HuBERT or traditional feature sets like OpenSMILE and MFCC. Visual encoding is typically handled by CNN architectures like ResNet and EfficientNet, with a growing adoption of vision Transformers. Fusion strategies vary considerably, from attention and MLPs to graph networks, yet all remain confined to single-task optimization without mechanisms for cross-task knowledge transfer. Critically, every method in the table operates under fully supervised learning within a single domain. In contrast, SCD-MMPSR is the first framework to enable semi-supervised, cross-domain, and multitask learning protocols across ER, PTR, and AHR, overcoming the annotation and generalization bottlenecks that constrain existing methods.

## A.2 PROPOSED PROMPT AND EXAMPLE OF BEHAVIOR DESCRIPTION

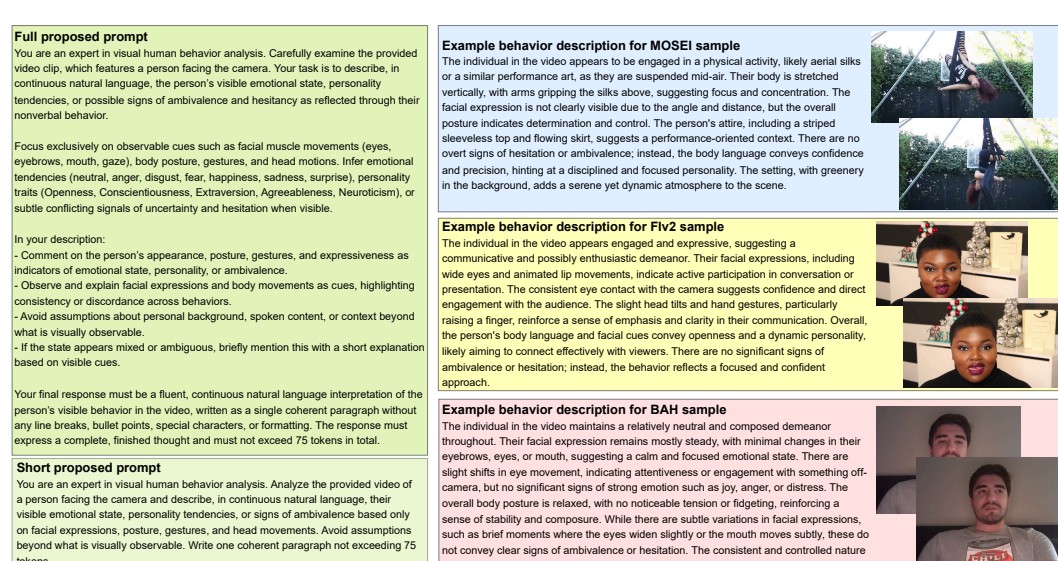

Figure 3: Proposed prompt and example of behavior description.

Table 3: Overview of SOTA methods

| Method | Linguistic Features | Acoustic Features | Visual Features | Modality Fusion | Task | Learning Type | Learning Domain |
|---|---|---|---|---|---|---|---|
| Joshi et al. (2022) | sBERT | openSMILE, CNN | OpenFace 2.0, Multi-Comp OpenFace | GNN | ER | Supervised | Single-domain |
| Goncalves et al. (2023) | – | Wav2Vec2-large-robust | EfficientNet-B2 | Cross-Modal Transformer | ER | Supervised | Single-domain |
| Li et al. (2023) | GloVe, BERT | COVAREP | Facet | Graph-based Knowledge Distillation | ER | Supervised | Single-domain |
| Deng et al. (2024) | Transformer | Transformer | – | Cross-Modal Attention, Multi-head Attention | ER | Supervised | Single-domain |
| Chandraumakantham et al. (2024) | DistilRoBERTa | openSMILE | PyFeat | LLM | ER | Supervised | Single-domain |
| Hosseini et al. (2024) | BiLSTM | CNN-LSTM | Inception-ResNet-v2 | DNN, decision-level fusion using regression softmax | ER | Supervised | Single-domain |
| Boitel et al. (2025) | DeBERTa | Semi-CNN | ResNet-50, 3D-CNN | MIST framework | ER | Supervised | Single-domain |
| Farhadipour et al. (2025) | RoBERTa | Wav2Vec2 | FacialNet, BiLSTM, CNN, Transformer | CNN, Transformer | ER | Supervised | Single-domain |
| Liu et al. (2025) | – | – | Spatio-Temporal vision Transformer | Dynamic Feature Fusion | ER | Supervised | Single-domain |
| Zhang et al. (2025a) | Deberta | openSMILE | DenseNet | Cross-modal Transformer, Mamba | ER | Supervised | Single-domain |
| Markitantov et al. (2025) | XLMRoBERTa, JINA | Wav2Vec2, ExHuBERT | YOLO, EmoAffect-Net, ResEmoteNet | BFS, LEFS, LEFSA | ER, SR | Supervised | Multi-domain |
| Zhao et al. (2023) | – | VGGish | VGG-Face | Decision-level fusion strategy | PTR | Supervised | Single-domain |
| Agrawal et al. (2023) | XLM-RoBERTa | Trill-Distilled | R(2+1)D, Video Swin Transformer | Fat Transformer Cross-Attention | PTR | Supervised | Single-domain |
| Yuanchao et al. (2023) | – | Transformer, Wav2Vec2 | – | – | PTR | Supervised | Single-domain |
| Ryumina et al. (2024) | BERT+BiLSTM, LIWC+ReBiLSTM | VGG-16+FCNN, openSMILE+LSTM | EmoAffectNet+LSTM, geometric features+LSTM | GSFN | PTR | Supervised | Cross-domain |
| Kong et al. (2025) | – | MFCC | EfficientFace | Feature concatenation, Attention Module | PTR | Supervised | Single-domain |
| Masumura et al. (2025) | BERT | HuBERT | CenterNet, MobileNetV3, Transformer, VGGFace2 | Transformer | PTR | Supervised | Single-domain |
| Bao et al. (2025) | RoBERTa | ResNet-34 | X3D, Temporal encoder | Transformer | PTR | Supervised | Single-domain |
| Hallmen et al. (2025) | Whisper, GTE-Large | Wav2Vec2 (with VAD) | ViT-Huge | MLP fusion, convolution-like temporal modeling | AHR | Supervised | Single-domain |
| Savchenko & Savchenko (2025) | RoBERTa (GoEmotions), Whisper | Wav2Vec2, HuBERT | EmotiEffLib | MLP classifiers, early fusion, blending, temporal smoothing | AHR | Supervised | Single-domain |
| Kollias et al. (2025) | BERT, TCN | VGGish, TCN | ResNet-50, TCN | Co-attention, classifier head | AHR | Supervised | Single-domain |
| SCD-MMPSR | EmoRoBERTa | CLAP | CLIP | Multimodal Cross-Domain Model | ER, PTR, AHR | Semi-supervised | Cross-domain |

Table 4: Comparison of prompt performance. Best and second-best results are highlighted

| Exp ID | Configuration | MOSEI | | FIv2 | | BAH | | Rank |
|---|---|---|---|---|---|---|---|---|
| | | mMF1 | mWACC | mACC | mCCC | MF1 | UAR | |
| 1 | Video+CLIP, Audio+CLAP, Text+CLAP, Behavior+CLIP (full proposed prompt) | 61.50 | 61.87 | 91.46 | 66.10 | 65.66 | 65.36 | 1.83 |
| 2 | Video+CLIP, Audio+CLAP, Text+CLAP, Behavior+CLIP (short proposed prompt) | 61.26 | 61.72 | 91.22 | 60.82 | 65.56 | 68.70 | 2.50 |
| 3 | Video+CLIP, Audio+CLAP, Text+CLAP, Behavior+CLIP ( Zhang et al. (2025b) prompt) | 60.87 | 61.52 | 90.79 | 61.94 | 66.78 | 66.40 | 2.67 |
| 4 | Video+CLIP, Audio+CLAP, Text+CLAP, Behavior+CLIP ( Cheng et al. (2024) prompt) | 60.36 | 61.28 | 91.10 | 61.29 | 66.52 | 67.98 | 3.00 |

Figure 3 and Table 4 present our prompt design for video-based behavior description and its impact on downstream performance in recognizing psychological states. The full proposed prompt (Exp-1) instructs the model to analyze visual behavior in a video clip, focusing on facial expressions, posture, gestures, and signs of ambivalence or hesitation, while avoiding assumptions about internal states. It emphasizes objective observation and fluent, continuous language output limited to 75 tokens. A shorter variant (Exp-2) retains core instructions but simplifies phrasing, leading to comparable or slightly improved results across all tasks. Both outperform existing baselines: the prompt from Zhang et al. (2025b) (Exp-3) and Cheng et al. (2024) (Exp-4), which were developed to analyze only human emotional states. These findings confirm that the proposed prompts, which focus on complex behavioral changes, improve the model's robustness and cover a broader range of psychological states in the video.

Table 5 compares the performance of our framework using different VLLMs for generating behavioral descriptions. Qwen2.5-VL-3B produces the best results across all three corpora and achieves the highest average rank (1.17), demonstrating its effectiveness in generating behaviorally informative textual summaries. InternVL2.5-4B, despite being larger (4B vs. 3B), performs competitively on MOSEI, but lags slightly on FIv2 and BAH. Eagle2-2B, the smallest model (2B parameters), exhibits noticeably lower performance measures, particularly on MOSEI and FIv2, suggesting that model capacity is crucial for capturing nuanced behavioral cues. These results suggest that the choice of VLLM has a significant impact on multimodal fusion performance.

Table 5: Experimental results of behavior encoders using different VLLM to describe behavior. Best and second-best results are highlighted

| Exp ID | Configuration | MOSEI | | FIv2 | | BAH | | Rank |
|---|---|---|---|---|---|---|---|---|
| | | mMF1 | mWACC | mACC | mCCC | MF1 | UAR | |
| 1 | Video+CLIP, Audio+CLAP, Text+CLAP, Behavior+CLIP using Qwen2.5-VL-3b (Bai et al., 2025) | 61.50 | 61.87 | 91.46 | 66.10 | 65.66 | 65.36 | 1.17 |
| 2 | Video+CLIP, Audio+CLAP, Text+CLAP, Behavior+CLIP using InternVL2.5-4b (Chen et al., 2024) | 61.99 | 59.75 | 91.04 | 64.96 | 63.68 | 63.76 | 2.17 |
| 3 | Video+CLIP, Audio+CLAP, Text+CLAP, Behavior+CLIP using Eagle2-2b (Chen et al., 2025) | 55.19 | 59.29 | 90.70 | 59.46 | 63.87 | 63.87 | 2.67 |

Table 6: Comparison of existing multimodal corpora

| Corpus | Conditions | Speech | Number of Records / Time | Task | Annotation protocol | Availability |
|---|---|---|---|---|---|---|
| IEMOCAP (Busso et al., 2008) | Laboratory | Spontaneous, pre-pared | 10039 utterances of 10 participants / 11.46 h | ER | Video, audio, motion capture of face, text; experts; seven emotions; one label | Open by request |
| MELD (Poria et al., 2019) | Movie scenes | Prepared | 1433 dialogs, 13708 utterances / 8h | ER | Video, audio, text; crowdsourcing; seven emotions; one label | Open |
| CMU-MOSEI (Zadeh et al., 2018b) | In-the-wild | Spontaneous | 3228 videos, 23453 utterances, about 1000 YouTube speaker / 65.88 h | ER | Video, audio, text; crowdsourcing; sentiment on a scale of $[-3, 3]$, six emotions on a scale of $[0, 3]$ annotated by multiple labels | Open |
| Aff-Wild2 (Kollias & Zafeiriou, 2019) | In-the-wild | Spontaneous | 558 videos, 458 participants / 43 h | ER | Video, audio; experts; valence and arousal (on a scale of $[-1, +1]$) and seven emotions annotated frame by frame | Open by request |
| MAHNOB-HCI (Wiem & Lachiri, 2017) | Laboratory | Spontaneous | 20 videos for 27 participants / 2 h | ER | EEG signal, audio, video, text; self-report; arousal and valence on a scale of 1-9; one label | Open |
| FIv2 (Escalante et al., 2020) | In-the-wild | Spontaneous | 10000 videos of about 3,000 participants/ 41 h | PTR | Video; crowdsourcing; Big Five on a scale of $[0, 1]$ | Open |
| UDIVA (Palmero et al., 2021) | Laboratory | Spontaneous | 188 sessions, 147 participants / 90.5 h | PTR | Audio, video, heart rate; self- and peer-reported; Big Five on a scale of $[-4, 4]$ | Open by request |
| MuPTA (Ryumina et al., 2023) | Laboratory | Spontaneous, pre-pared | 3870 videos, 30 participants / 7 h | PTR | Video, audio; self evaluation; Big Five on a scale of $[0, 1]$ | Open by request |
| BAH (González-González et al., 2025) | In-the-wild | spontaneous | 1118 videos, 224 participants / 8.26 h | AHR | Video, audio, text; experts; binary ambivalence and hesitancy | Open by request |

## A.3 COMPARISON OF EXISTING MULTIMODAL CORPORA

Table 6 provides an overview of existing multimodal corpora, comparing them along key characteristics: recording conditions, speech type, scale, target tasks, annotation protocol, and availability. For our study, we restrict training data to in-the-wild corpora: CMU-MOSEI for ER, FIv2 PTR, and BAH for AHR. This choice aligns with our focus on real-world applicability.

To evaluate cross-corpus generalization in ER, we use MELD, which provides audio, video, and text modalities and is annotated with the same seven emotions as CMU-MOSEI. This sets it apart from Aff-Wild2, which primarily consists of facial reactions to movie clips and often lacks informative audio or spoken content. IEMOCAP employs a distinct emotion label set and was recorded in controlled laboratory settings, whereas MAHNOB-HCI focuses on valence-arousal dimensions and is also based in the laboratory. For PTR, both MuPTA and UDIVA rely on self-evaluation of the BigFive traits under controlled laboratory conditions, which does not reflect the FIv2 corpus. Finally, BAH is the only multimodal corpus that targets ambivalence and hesitation, making it uniquely suitable for AHR.

## A.4 CLASSES DISTRIBUTIONS IN RESEARCH CORPORA

Figures 4 and 5 illustrate the class distributions across Train, Development, and Test subsets for the four research corpora used in our experiments: MOSEI, BAH, MELD, and FIv2.

Figure 4 shows that MOSEI exhibits a strong imbalance in emotion labels, with Happiness dominating the corpus (over 12,000 examples), while emotions such as Fear and Surprise are significantly underrepresented. The BAH corpus presents a balanced distribution of ambivalence classes, Absence and Presence, across all subsets, ensuring fair evaluation of AHR. The corpus BAH represents a nearly balanced distribution of ambivalence across all subsets, with a slight bias towards the Presence class. This ensures a fair estimation of AHR. The emotion distribution in MELD is unbalanced, with over 1,200 examples belonging to the Neutral class, while there are fewer than 100 examples for Fear and Disgust.

Figure 5 reveals that PTs scores follow continuous distributions across five Big Five dimensions: Openness, Conscientiousness, Extraversion, Agreeableness, and non-Neuroticism. Notably, most

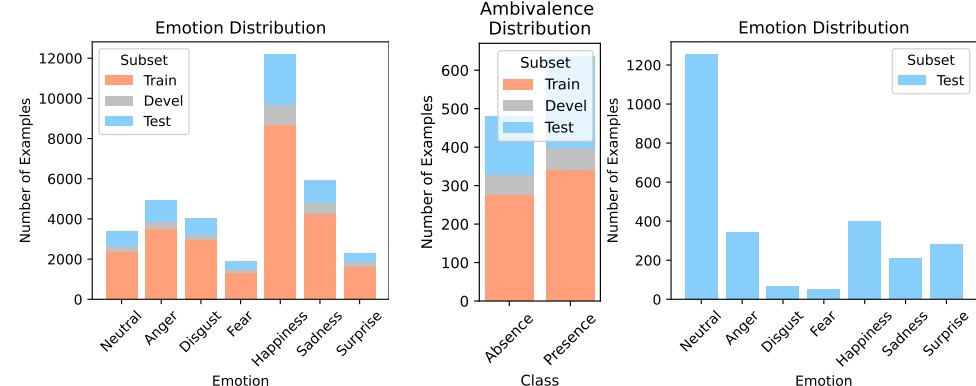

Figure 4: Distributions of classes in videos across subsets of MOSEI (left sub-figure), BAH (central sub-figure), and MELD (right sub-figure).

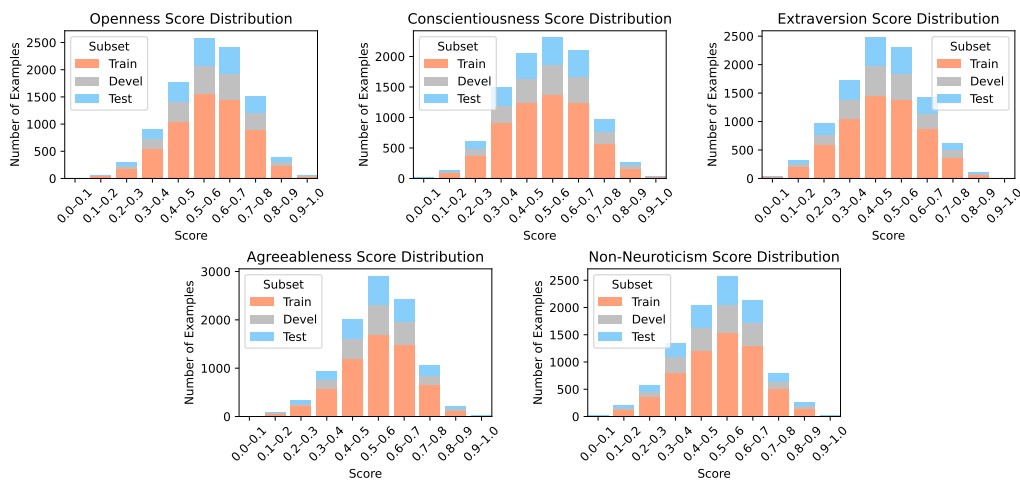

Figure 5: Distributions of PTs scores in videos across subsets of FIv2.

scores cluster in the mid-range (0.4–0.7), indicating a balanced representation of traits without extreme bias.

These distributions confirm that our experimental setup accounts for both categorical imbalances (MOSEI, BAH, and MELD) and continuous score variations (FIv2), enabling comprehensive evaluation of SCD-MMPSR's performance under realistic, heterogeneous conditions.

### A.5 COMPARATIVE ANALYSIS OF ENCODER PERFORMANCE

**Video encoders**. Table 7 evaluates eight visual encoders within the SCD-MMPSR framework under fixed audio, text, and behavior modalities. CLIP (Radford et al., 2021)[2], trained on image-caption pairs via contrastive learning, provides strong general-purpose visual representations. Google ViT (Dosovitskiy et al., 2021)[3], pre-trained on ImageNet for generic image classification. ResNet-50 (He et al., 2016)[4], a CNN backbone pre-trained on ImageNet. DinoV2 Large (Oquab et al., 2024)[5], a self-supervised vision Transformer trained without labels, provides robust generic features but lacks affective grounding. EmoViT v1[6], adapted for static facial ER, shows task-specific gains.

---

[2]https://huggingface.co/openai/clip-vit-base-patch32
[3]https://huggingface.co/google/vit-base-patch16-224
[4]https://huggingface.co/microsoft/resnet-50
[5]https://huggingface.co/facebook/dinov2-large
[6]https://huggingface.co/trpakov/vit-face-expression

Table 7: Experimental results of video encoders. Best and second-best results are highlighted

| Exp ID | Configuration | MOSEI | | FIv2 | | BAH | | Rank |
|---|---|---|---|---|---|---|---|---|
| | | mMF1 | mWACC | mACC | mCCC | MF1 | UAR | |
| 1 | Video+CLIP (Radford et al., 2021) (30 frames), Audio+CLAP, Text+CLAP, Behavior+CLIP | 61.50 | 61.87 | 91.46 | 66.10 | 65.66 | 65.36 | 5.50 |
| 2 | Video+CLIP (Radford et al., 2021) (20 frames), Audio+CLAP, Text+CLAP, Behavior+CLIP | 61.27 | 62.15 | 90.79 | 61.94 | 66.78 | 66.40 | 6.17 |
| 3 | Video+CLIP (Radford et al., 2021) (40 frames), Audio+CLAP, Text+CLAP, Behavior+CLIP | 61.98 | 62.62 | 91.20 | 63.13 | 64.27 | 65.01 | 5.67 |
| 4 | Video+Google ViT (Dosovitskiy et al., 2021) (30 frames), Audio+CLAP, Text+CLAP, Behavior+CLIP | 61.29 | 61.91 | 91.17 | 62.67 | 66.61 | 66.53 | 5.83 |
| 5 | Video+ResNet-50 (He et al., 2016) (30 frames), Audio+CLAP, Text+CLAP, Behavior+CLIP | 56.84 | 59.52 | 89.92 | 50.51 | 68.26 | 68.55 | 7.00 |
| 6 | Video+DinoV2 Large (Oquab et al., 2024) (30 frames), Audio+CLAP, Text+CLAP, Behavior+CLIP | 61.65 | 62.03 | 91.31 | 64.24 | 66.66 | 66.51 | 4.17 |
| 7 | Video+EmoViT v1 (30 frames), Audio+CLAP, Text+CLAP, Behavior+CLIP | 61.08 | 62.29 | 90.49 | 58.28 | 67.43 | 67.25 | 5.67 |
| 8 | Video+EmoViT v2 (30 frames), Audio+CLAP, Text+CLAP, Behavior+CLIP | 61.39 | 61.56 | 91.11 | 62.90 | 65.98 | 65.88 | 7.00 |
| 9 | Video+EmoAffectNet (Ryumina et al., 2022) (30 frames), Audio+CLAP, Text+CLAP, Behavior+CLIP | 62.07 | 62.69 | 90.68 | 56.69 | 67.30 | 66.99 | 4.50 |
| 10 | Video+EmotiEffLib (Savchenko, 2023) (30 frames), Audio+CLAP, Text+CLAP, Behavior+CLIP | 62.57 | 62.73 | 91.29 | 65.48 | 66.21 | 66.09 | 3.50 |

Table 8: Experimental results of audio encoders. Best and second-best results are highlighted

| Exp ID | Configuration | MOSEI | | FIv2 | | BAH | | Rank |
|---|---|---|---|---|---|---|---|---|
| | | mMF1 | mWACC | mACC | mCCC | MF1 | UAR | |
| 1 | Video+CLIP, Audio+CLAP (Wu et al., 2023), Text+CLAP, Behavior+CLIP | 61.50 | 61.87 | 91.46 | 66.10 | 65.66 | 65.36 | 3.50 |
| 2 | Video+CLIP, Audio+Whisper-base (Radford et al., 2023), Text+CLAP, Behavior+CLIP | 57.98 | 60.34 | 90.84 | 62.52 | 67.39 | 67.83 | 5.00 |
| 3 | Video+CLIP, Audio+AST (Gong et al., 2021), Text+CLAP, Behavior+CLIP | 60.78 | 62.58 | 91.08 | 60.75 | 66.62 | 66.36 | 3.67 |
| 4 | Video+CLIP, Audio+Wav2Vec2 (Baevski et al., 2020), Text+CLAP, Behavior+CLIP | 61.45 | 62.04 | 91.06 | 63.40 | 65.22 | 65.56 | 3.83 |
| 5 | Video+CLIP, Audio+EmoWav2Vec2 (Wagner et al., 2023), Text+CLAP, Behavior+CLIP | 61.27 | 63.13 | 91.43 | 65.11 | 67.91 | 67.96 | 1.83 |
| 6 | Video+CLIP, Audio+EmoExHuBERT (Amiriparian et al., 2024), Text+CLAP, Behavior+CLIP | 58.39 | 61.60 | 90.86 | 55.93 | 67.18 | 69.11 | 4.83 |

EmoViT v2[7], optimized for ER in images. Both the VIT-based models were fine-tuned using the FER2013 corpus. EmoAffectNet (Ryumina et al., 2022)[8], based on ResNet-50, and was fine-tuned with different augmentation techniques on AffectNet for in-the-wild ER. EmotiEffLib (Savchenko, 2023)[9], a lightweight library optimized for real-time facial affect analysis in video sequences. This model achieves the highest overall rank by effectively capturing dynamic, context-aware facial cues across ER, PTR, and AHR. Evaluation of SCD-MMPSR under varying numbers of uniformly sampled frames (Exp 1-3) reveals that 30 frames yield optimal performance.

**Audio encoders**. Table 8 compares six audio encoders within the SCD-MMPSR framework under identical multimodal conditions. CLAP (Wu et al., 2023)[10], trained on large-scale audio-text pairs with contrastive learning, aligns audio representations with semantic textual descriptions. Whisper-base (Radford et al., 2023)[11], trained for multilingual speech recognition and translation, offers robustness to noise and accents but is optimized for lexical content rather than paralinguistic cues. Audio Spectrogram Transformer (AST) (Gong et al., 2021)[12], a spectrogram-based Transformer pre-trained on AudioSet for environmental sound classification, proves less suitable for vocal affect due to its domain mismatch. Wav2Vec2 (Baevski et al., 2020)[13], fine-tuned for phonetic recognition on LibriSpeech, captures linguistic structure effectively but lacks explicit modeling of emotional prosody. EmoWav2Vec2 (Wagner et al., 2023)[14], a Wav2Vec2 variant fine-tuned on MSP-Podcast to predict arousal, dominance, and valence, provides both dimensional emotion logits and affect-rich pooled hidden states from its last transformer layer. EmoExHuBERT (Amiriparian et al., 2024)[15], an extension of HuBERT fine-tuned on multiple emotion corpora, is explicitly designed to extract expressive paralinguistic features and predict dimensional affect. Results show that EmoWav2Vec2 achieves the best overall performance. This confirms that encoders explicitly optimized for affective representation deliver superior transferability for psychological state recognition tasks compared to general-purpose speech, environmental audio, or even contrastively aligned models like CLAP.

**Text / behavior encoders**. Tables 9 and 10 compares eight encoders for text and behavioral modalities within the SCD-MMPSR framework under identical multimodal conditions. CLAP (Wu et al.,

---

[7]https://huggingface.co/dima806/facial_emotions_image_detection
[8]https://github.com/ElenaRyumina/EMO-AffectNetModel
[9]https://github.com/sb-ai-lab/EmotiEffLib
[10]https://huggingface.co/laion/clap-htsat-fused
[11]https://huggingface.co/openai/whisper-base
[12]https://huggingface.co/MIT/ast-finetuned-audioset-10-10-0.4593
[13]https://huggingface.co/facebook/wav2vec2-base-960h
[14]https://huggingface.co/audeering/wav2vec2-large-robust-12-ft-emotion-msp-dim
[15]https://huggingface.co/amiriparian/ExHuBERT

Table 9: Experimental results of text encoders. Best and second-best results are highlighted

| Exp ID | Configuration | MOSEI | | FIv2 | | BAH | | Rank |
|---|---|---|---|---|---|---|---|---|
| | | mMF1 | mWACC | mACC | mCCC | MF1 | UAR | |
| 1 | Video+CLIP, Audio+CLAP, Text+CLAP (Wu et al., 2023), Behavior+CLIP | 61.50 | 61.87 | 91.46 | 66.10 | 65.66 | 65.36 | 6.33 |
| 2 | Video+CLIP, Audio+CLAP, Text+JinaV3 (Sturua et al., 2024), Behavior+CLIP | 63.57 | 65.18 | 91.30 | 66.07 | 66.76 | 68.19 | 4.50 |
| 3 | Video+CLIP, Audio+CLAP, Text+JinaV4 (Günther et al., 2025), Behavior+CLIP | 58.28 | 69.36 | 91.84 | 69.18 | 63.27 | 63.37 | 4.50 |
| 4 | Video+CLIP, Audio+CLAP, Text+BGE (Xiao et al., 2024a), Behavior+CLIP | 63.73 | 64.95 | 91.36 | 63.93 | 68.22 | 68.14 | 4.67 |
| 5 | Video+CLIP, Audio+CLAP, Text+RoBERTa (Liu et al., 2019), Behavior+CLIP | 63.39 | 64.33 | 91.44 | 65.44 | 68.76 | 68.68 | 3.83 |
| 6 | Video+CLIP, Audio+CLAP, Text+XLM RoBERTa (Conneau et al., 2019), Behavior+CLIP | 62.93 | 63.59 | 91.70 | 68.25 | 67.56 | 67.92 | 4.66 |
| 7 | Video+CLIP, Audio+CLAP, Text+EmoDistilRoBERTa (Sanh et al., 2019), Behavior+CLIP | 63.05 | 64.15 | 91.53 | 66.33 | 68.68 | 68.56 | 3.83 |
| 8 | Video+CLIP, Audio+CLAP, Text+EmoRoBERTa, Behavior+CLIP | 63.10 | 64.77 | 91.57 | 66.07 | 68.61 | 68.59 | 3.50 |

Table 10: Experimental results of behavior encoders by text. Best and second-best results are highlighted

| Exp ID | Configuration | MOSEI | | FIv2 | | BAH | | Rank |
|---|---|---|---|---|---|---|---|---|
| | | mMF1 | mWACC | mACC | mCCC | MF1 | UAR | |
| 1 | Video+CLIP, Audio+CLAP, Text+CLAP, Behavior+CLIP (Radford et al., 2021) | 61.50 | 61.87 | 91.46 | 66.10 | 65.66 | 65.36 | 4.67 |
| 2 | Video+CLIP, Audio+CLAP, Text+CLAP, Behavior+JinaV3 (Sturua et al., 2024) | 61.71 | 62.89 | 91.21 | 63.85 | 63.95 | 63.83 | 5.33 |
| 3 | Video+CLIP, Audio+CLAP, Text+CLAP, Behavior+JinaV4 (Günther et al., 2025) | 59.30 | 61.24 | 91.33 | 63.34 | 66.65 | 66.34 | 6.33 |
| 4 | Video+CLIP, Audio+CLAP, Text+CLAP, Behavior+BGE (Xiao et al., 2024a) | 59.03 | 61.88 | 91.16 | 62.13 | 66.69 | 67.96 | 5.83 |
| 5 | Video+CLIP, Audio+CLAP, Text+CLAP, Behavior+RoBERTa (Liu et al., 2019) | 60.51 | 62.23 | 91.36 | 65.35 | 67.43 | 67.25 | 3.83 |
| 6 | Video+CLIP, Audio+CLAP, Text+CLAP, Behavior+XLM RoBERTa (Conneau et al., 2019) | 61.32 | 62.68 | 91.34 | 63.45 | 67.37 | 67.28 | 4.00 |
| 7 | Video+CLIP, Audio+CLAP, Text+CLAP, Behavior+EmoDistilRoBERTa (Sanh et al., 2019) | 61.96 | 62.30 | 91.45 | 65.05 | 65.67 | 65.55 | 3.83 |
| 8 | Video+CLIP, Audio+CLAP, Text+CLAP, Behavior+EmoRoBERTa | 61.53 | 62.92 | 91.40 | 67.17 | 67.15 | 67.62 | 2.17 |

2023), a contrastive audio-language model, aligns textual representations with acoustic semantics but is not optimized for psychological nuance. CLIP (Radford et al., 2021), a contrastive vision-language model, captures general semantic grounding but lacks specialization for affective or behavioral cues. JinaV3 (Sturua et al., 2024)[16], a 570M-parameter multilingual transformer with LoRA adapters, supports long contexts (8192 tokens) and excels in retrieval but is not fine-tuned for psychological states recognition. JinaV4 (Günther et al., 2025)[17], a 3.8B-parameter multimodal encoder based on Qwen2.5-VL-3b, unifies text and image representations. BGE (Xiao et al., 2024a)[18], a BERT-based dense retriever, is highly effective for semantic matching and classification but lacks dialogue-aware or affective tuning. RoBERTa (Liu et al., 2019)[19], trained on 160GB of English text with dynamic masking, offers strong general-purpose contextual embeddings but is not emotion-specialized. XLM RoBERTa (Conneau et al., 2019)[20], pre-trained on 100 languages, provides robust cross-lingual features but similarly lacks affective grounding. EmoDistilRoBERTa (Sanh et al., 2019)[21], a distilled model fine-tuned on multi-domain emotion corpora (Twitter, Reddit, etc.), is lightweight and efficient for ER. EmoRoBERTa[22] is a version of EmoDistilRoBERTa fine-tuned on transcripts from multiple corpora (Crowdflower, GoEmotions, etc.) for ER. Across both text and behavior modalities, EmoRoBERTa outperforms all alternatives. Unlike general-purpose encoders (CLAP, BGE, RoBERTa) or multilingual/retrieval models (JinaV3/V4, XLM RoBERTa), EmoRoBERTa is fine-tuned specifically on emotionally annotated dialogue.

We conduct additional experiments to assess whether behavior representations can be effectively derived without VLLMs, using only lightweight visual encoders and scene-level visual context (as opposed to text-based behavioral descriptions). As shown in Table 11, behavior encoding based on scene analysis outperforms text-based encoding across all three corpora. This indicates that holistic visual context often provides more stable and informative behavioral cues than current VLLM-generated summaries. We nevertheless adopt VLLM-generated textual behavior descriptions because this approach is novel, interpretable, and enables semantic reasoning, offering a path toward human-aligned, language-mediated analysis that pure visual features cannot provide.

---

[16] https://huggingface.co/jinaai/jina-embeddings-v3
[17] https://huggingface.co/jinaai/jina-embeddings-v4
[18] https://huggingface.co/BAAI/bge-large-en
[19] https://huggingface.1319lm.top/FacebookAI/roberta-large
[20] https://huggingface.co/FacebookAI/xlm-roberta-large
[21] https://huggingface.co/j-hartmann/emotion-english-distilroberta-base
[22] https://huggingface.co/michellejieli/emotion_text_classifier

Table 11: Experimental results of behavior encoders by scene. Best and second-best results are highlighted

| Exp ID | Configuration | MOSEI | | FIv2 | | BAH | | Rank |
|---|---|---|---|---|---|---|---|---|
| | | mMF1 | mWACC | mACC | mCCC | MF1 | UAR | |
| 1 | Video+CLIP, Audio+CLAP, Text+CLAP, Behavior+CLIP (Radford et al., 2021) by text | 61.50 | 61.87 | 91.46 | 66.10 | 65.66 | 65.36 | 3.67 |
| 2 | Video+CLIP, Audio+CLAP, Text+CLAP, Behavior+CLIP (Radford et al., 2021) by scene | 61.53 | 62.36 | 91.58 | 69.80 | 68.61 | 68.59 | 1.17 |
| 3 | Video+CLIP, Audio+CLAP, Text+CLAP, Behavior+Google ViT (Dosovitskiy et al., 2021) by scene | 60.18 | 62.14 | 90.93 | 64.83 | 68.13 | 67.68 | 3.50 |
| 4 | Video+CLIP, Audio+CLAP, Text+CLAP, Behavior+ResNet-50 (He et al., 2016) by scene | 59.28 | 61.24 | 91.31 | 65.28 | 67.67 | 67.19 | 4.17 |
| 5 | Video+CLIP, Audio+CLAP, Text+CLAP, Behavior+DinoV2 Large (Oquab et al., 2024) by scene | 61.50 | 62.60 | 91.52 | 67.66 | 67.15 | 67.62 | 2.33 |

Table 12: Experimental results of various combinations of modality encoders. Best and second-best results are highlighted

| Exp ID | Configuration | MOSEI | | FIv2 | | BAH | | Rank |
|---|---|---|---|---|---|---|---|---|
| | | mMF1 | mWACC | mACC | mCCC | MF1 | UAR | |
| 1 | Video+CLIP, Audio+CLAP, Text+CLAP, Behavior+CLIP | 61.50 | 61.87 | 91.46 | 66.10 | 65.66 | 65.36 | 6.00 |
| 2 | Video+EmotiEffLib, Audio+CLAP, Text+CLAP, Behavior+CLIP | 62.57 | 62.73 | 91.29 | 65.48 | 66.21 | 66.09 | 6.50 |
| 3 | Video+CLIP, Audio+EmoWav2Vec2, Text+CLAP, Behavior+CLIP | 61.27 | 63.13 | 91.43 | 65.11 | 67.91 | 67.96 | 5.33 |
| 4 | Video+CLIP, Audio+CLAP, Text+EmoRoBERTa, Behavior+CLIP | 63.10 | 64.77 | 91.57 | 66.07 | 68.61 | 68.59 | 2.50 |
| 5 | Video+CLIP, Audio+CLAP, Text+CLAP, Behavior+EmoRoBERTa | 61.53 | 62.92 | 91.40 | 67.17 | 67.15 | 67.62 | 4.67 |
| 6 | Video+EmotiEffLib, Audio+Wav2vec, Text+EmoRoBERTa, Behavior+EmoRoBERTa | 63.55 | 63.75 | 91.30 | 65.33 | 67.12 | 66.93 | 5.00 |
| 7 | Video+CLIP, Audio+Wav2vec, Text+EmoRoBERTa, Behavior+EmoRoBERTa | 63.36 | 64.41 | 91.40 | 62.99 | 68.00 | 69.35 | 3.67 |
| 8 | Video+CLIP, Audio+CLAP, Text+EmoRoBERTa, Behavior+EmoRoBERTa | 63.40 | 64.00 | 91.44 | 66.68 | 69.29 | 69.07 | 2.17 |

Table 12 summarizes the performance of various multimodal configurations that combine the top-performing unimodal encoders from prior ablation studies. We selected the strongest candidates for each modality (EmotiEffLib for video, EmoWav2Vec2 for audio, and EmoRoBERTa for text and behavior) and fused them to evaluate their combined contribution. The results show that the optimal configuration is Video+CLIP, Audio+CLAP, Text+EmoRoBERTa, and Behavior+EmoRoBERTa (ID-8). This combination achieves the highest overall rank (2.17), as well as top scores on BAH (MF1: 69.29, UAR: 69.07), and a strong performance on MOSEI and FIv2. This configuration demonstrates that using affect-specialized encoders for text and behavior (EmoRoBERTa) provides greater gains than modality-specific models for visual or acoustic data, even when combined with general-purpose models such as CLIP and CLAP. Replacing CLAP with EmoWav2Vec2 (ID-3) or CLIP with EmotiEffLib (ID-2) results in marginal or inconsistent improvements. This suggests that linguistic modeling of psychological states is the primary driver of cross-task generalization in our framework.

### A.6 COMPARATIVE ANALYSIS OF GRAPH LAYERS AND ATTENTION MECHANISM

**Attention mechanisms.** Table 13 compares four advanced attention variants with the vanilla Multi-Head Attention (MHA) mechanism (Vaswani et al., 2017). Multi-Token Attention (MTA) (Golovneva et al., 2025)[23] conditions attention weights on multiple query and key vectors at once. Within each head, this mechanism applies a convolution operation to attention scores using both a key-query and head convolution, repeating the process after softmax and adding a scalar gating function before final concatenation. This allows for fine-grained, multi-scale interaction modeling. Cross-attention Message-Passing Transformer (CrossMPT) (Park et al., 2025)[24] uses two cross-attention blocks to iteratively update query and key-value representations, improving multimodal alignment through iterative refinement. Bidirectional Cross Attention (BiCA) (Hiller et al., 2024)[25] allows input tokens and latent variables to attend to each other simultaneously. It leverages emergent attention symmetry for balanced bidirectional information flow. Forgetting Attention (FA) (Lin et al., 2025)[26] introduces a forget gate within the softmax attention mechanism. This gate down-weights unnormalized attention scores in a data-dependent manner, mimicking cognitive filtering of irrelevant signals. The comparison results show that MTA achieves the best overall rank (2.33), particularly excelling on FIv2 and BAH. This confirms that its convolution-augmented, multi-stage normalization architecture better captures cross-modal psychological dependencies than iterative, symmetric, or gating-based mechanisms.

---

[23]https://github.com/facebookresearch/RAM/tree/main/projects/mta
[24]https://github.com/iil-postech/crossmpt
[25]https://github.com/lucidrains/bidirectional-cross-attention
[26]https://github.com/zhixuan-lin/forgetting-transformer/tree/main

Table 13: Experimental results on attention mechanisms. Best and second-best results are highlighted

| Exp ID | Configuration | MOSEI | | FIv2 | | BAH | | Rank |
|---|---|---|---|---|---|---|---|---|
| | | mMF1 | mWACC | mACC | mCCC | MF1 | UAR | |
| 1 | Video+CLIP, Audio+CLAP, Text+CLAP, Behavior+CLIP, MHA (Vaswani et al., 2017) | 61.50 | 61.87 | 91.46 | 66.10 | 65.66 | 65.36 | 3.67 |
| 2 | Video+CLIP, Audio+CLAP, Text+CLAP, Behavior+CLIP, MTA (Golovneva et al., 2025) | 60.83 | 61.45 | 91.59 | 68.04 | 68.76 | 68.36 | 2.33 |
| 3 | Video+CLIP, Audio+CLAP, Text+CLAP, Behavior+CLIP, CrossMPT (Park et al., 2025) | 61.56 | 62.60 | 91.32 | 65.06 | 68.75 | 68.82 | 2.83 |
| 4 | Video+CLIP, Audio+CLAP, Text+CLAP, Behavior+CLIP, BiCA (Hiller et al., 2024) | 61.81 | 62.49 | 91.50 | 67.82 | 66.29 | 67.28 | 2.50 |
| 5 | Video+CLIP, Audio+CLAP, Text+CLAP, Behavior+CLIP, FA (Lin et al., 2025) | 60.10 | 61.06 | 91.33 | 65.61 | 68.12 | 69.21 | 3.67 |

Table 14: Experimental results on GNNs. Best and second-best results are highlighted

| Exp ID | Configuration | MOSEI | | FIv2 | | BAH | | Rank |
|---|---|---|---|---|---|---|---|---|
| | | mMF1 | mWACC | mACC | mCCC | MF1 | UAR | |
| 1 | Video+CLIP, Audio+CLAP, Text+CLAP, Behavior+CLIP, vanilla GNN (Veličković et al., 2018) | 61.50 | 61.87 | 91.46 | 66.10 | 65.66 | 65.36 | 4.17 |
| 2 | Video+CLIP, Audio+CLAP, Text+CLAP, Behavior+CLIP, NCGNN (Wang & Cho, 2024) | 61.51 | 63.31 | 91.40 | 65.33 | 66.99 | 67.12 | 2.67 |
| 3 | Video+CLIP, Audio+CLAP, Text+CLAP, Behavior+CLIP, EDGNN (Pahng & Hormoz, 2025) | 61.84 | 61.89 | 91.62 | 68.99 | 66.00 | 66.17 | 2.67 |
| 4 | Video+CLIP, Audio+CLAP, Text+CLAP, Behavior+CLIP, UCGNN (Kiani et al., 2024) | 62.43 | 63.22 | 91.74 | 70.14 | 65.47 | 65.49 | 2.33 |
| 5 | Video+CLIP, Audio+CLAP, Text+CLAP, Behavior+CLIP, HGNN (Yue et al., 2025) | 61.48 | 62.81 | 91.35 | 67.42 | 66.89 | 68.05 | 3.17 |

**Graph Neural Network.** Table 14 compares four GNN variants with the vanilla one (Veličković et al., 2018). Non-Convolutional GNN (NCGNN) (Wang & Cho, 2024)[27] replaces conventional message passing with random walks guided by a unified memory. This GNN utilizes RNN to aggregate topological and semantic signals along node-anchored walks, thereby mitigating the limitations of expressiveness and over-smoothing without relying on sparse convolutional kernels. Edge Directions GNN (EDGNN) (Pahng & Hormoz, 2025)[28] introduces learnable edge directions, encoded in a complex-valued Laplacian. The real and imaginary parts of the Laplacian encode opposite information flows. Messages from in- and out-neighbors are combined with optional self-features to enable differentiable, long-range directional propagation on directed and undirected graphs. Unitary Convolutions GNN (UCGNN) (Kiani et al., 2024)[29] stabilize deep graph models by enforcing unit-modulus transformations that avoid over-smoothing and improve training stability as the depth increases. Hyperbolic GNN (HGNN) (Yue et al., 2025)[30] recasts message passing as a system of hyperbolic partial differential equations. This method offers spectral-spatiotemporal interpretability and enhanced performance by evolving node states in a solution space spanned by Laplacian eigenvectors. Results show that UCGNN achieves the best overall rank (2.33), confirming that depth-stable, unit-modulus architectures are critical for modeling complex, cross-task psychological states interactions in graph-based fusion.

## A.7 OPTIMIZATION OF MODEL AND TRAINING HYPERPARAMETERS

To optimize performance across multimodal corpora, a comprehensive grid search was conducted over key training and model hyperparameters. Starting from a strong baseline Exp-3 (see Table 1), we explored variations in hidden states (hidden_dim) and output feature (out_features) dimensions, transformer head count (num_transformer_heads), dropout rate (dropout), learning rate (lr), scheduler type (scheduler_type), and optimizer choice. The search results are presented in Table 15. All experiments presented in Appendix A.2, A.5 and A.6 are carried out under the baseline values of hyperparameters, while the task contribution coefficients ($w_t^s$, $w_t^{ss}$) are fixed at 1.0.

The search revealed that increasing model capacity via hidden_dim and out_features to 512 consistently improved generalization without overfitting, particularly benefiting two corpora (FIv2 and BAH). A moderate dropout of 0.15 offered the best regularization, while the `plateau` scheduler proved most effective in stabilizing late-stage training by adapting to loss plateaus. The remaining parameters remained unchanged due to the search and showed no improvement.

---

[27]https://github.com/ak24watch/RUM-Graph-nets/tree/main

[28]https://github.com/hormoz-lab/coed-gnn/tree/main

[29]https://github.com/Weber-GeoML/Unitary_Convolutions/tree/main

[30]https://github.com/YueAWu/Hyperbolic-GNN/tree/main

Table 15: Grid search result of hyperparameters

| Hyperparameter | Baseline value | Search values | Best value |
|---|---|---|---|
| hidden_dim | 256 | [128, 256, 512, 1024] | 512 |
| out_features | 256 | [128, 256, 512, 1024] | 512 |
| num_transformer_heads | 8 | [2, 4, 8, 16] | 8 |
| dropout | 0.2 | [0.0, 0.1, 0.15, 0.2, 0.25, 0.3] | 0.15 |
| scheduler_type | none | [none, plateau, cosine, onecycle] | plateau |
| lr | $10^{-4}$ | $[10^{-3}, 10^{-4}, 10^{-5}]$ | $10^{-4}$ |
| optimizer | adam | [adam, adamw, lion, sgd, rmsprop] | adam |

Applying the best-performing hyperparameters (Exp-5, as shown in Table 1) resulted in a performance improvement. The gains were most significant on FIv2 and BAH, where classification and regression measures improved, indicating increased robustness to cross-task variability. MOSEI metrics decreased, suggesting either saturation of this corpus or a need for task-specific fine-tuning. Overall, these results indicate that careful parameter selection can lead to improved model performance.

## A.8 OPTIMIZATION OF SEMI-SUPERVISED LEARNING HYPERPARAMETERS

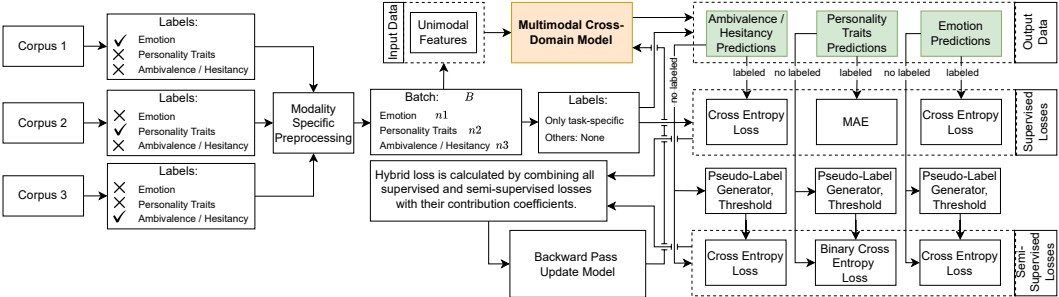

Figure 6: Training process pipeline.

Figure 6 shows the training process of SCD-MMPSR, a framework that can recognize three psychological states simultaneously. The framework uses three different corpora (MOSEI, FIv2, and BAH) that are annotated for various tasks. These corpora are used to extract features for each modality, which are then fed into a MCDM. MCDM generates predictions for all three tasks at once. A hybrid loss function is employed, combining supervised losses and semi-supervised losses. This allows the model to learn from both labeled and unlabeled data. Pseudo labels are generated based on confidence thresholds and are updated during the training process. This helps the model learn robust representations for all tasks without requiring co-annotated data.

The SSL hyperparameters are optimized using grid search. The results are presented in Table 16. This search reveals that optimal pseudo-labeling requires lower confidence thresholds (0.60) than commonly assumed. It indicates that moderately confident predictions contain a valuable signal for cross-task learning. The gradient balancing controllers have approximately equal values ($\alpha^{\mathrm{s}} = 1.25$ and $\alpha^{\mathrm{ss}} = 1.0$). This confirms that unlabeled data contributes substantially, but only when properly scaled. Learning rates ($\eta_w^{\mathrm{s}} = 0.01$ and $\eta_w^{\mathrm{ss}} = 0.005$) are best set higher than the baseline for both loss types, suggesting that a faster adaptation improves convergence. The budget coefficient peaks at $\lambda = 0.3$. This suggests that 30% of training steps should be devoted to pseudo-label refinement to maximize gain. Finally, preserving a minimal task contribution of $w_{\mathrm{floor}} = 10^{-3}$ prevents gradient starvation for weaker tasks. Together, these settings create a best-performing SSL: lower thresholds, higher semi-supervised weights, aggressive learning, and controlled budgeting, unlocking the full potential of unlabeled data in cross-domain multitask learning.

Figure 7 shows the adaptive change in task contribution coefficients for each epoch with the best hyperparameters SSL. The contribution coefficients for all tasks are dynamically adjusted at each

Table 16: Grid search result of SSL hyperparameters

| Hyperparameter | Baseline value | Search values | Best value |
|---|---|---|---|
| Pseudo-label threshold $\tau_{EMO/AH}$ | 0.8 | [0.5, 0.6, 0.7, 0.8, 0.9] | 0.6 |
| Pseudo-label threshold $\tau_{PT}$ | 0.5 | [0.5, 0.55, 0.6, 0.56] | 0.6 |
| Gradient balancing controller $\alpha^{\text{s}}$ | 1 | [1.0, 1.25, 1.50, 1.75] | 1.25 |
| Gradient balancing controller $\alpha^{\text{ss}}$ | 0.25 | [0.25, 0.50, 0.75, 1.0, 1.25] | 1.0 |
| Learning rate $\eta_w^{\text{s}}$ | 0.005 | [0.005, 0.01, 0.025] | 0.01 |
| Learning rate $\eta_w^{\text{ss}}$ | 0.004 | [0.004, 0.005, 0.006] | 0.005 |
| Budget coefficient $\lambda$ | 0.1 | [0.1, 0.2, 0.3, 0.4] | 0.3 |
| Min value of the task contribution coefficients $w_{\text{floor}}$ | $10^{-3}$ | $[10^{-2}, 10^{-3}, 10^{-4}]$ | $10^{-3}$ |

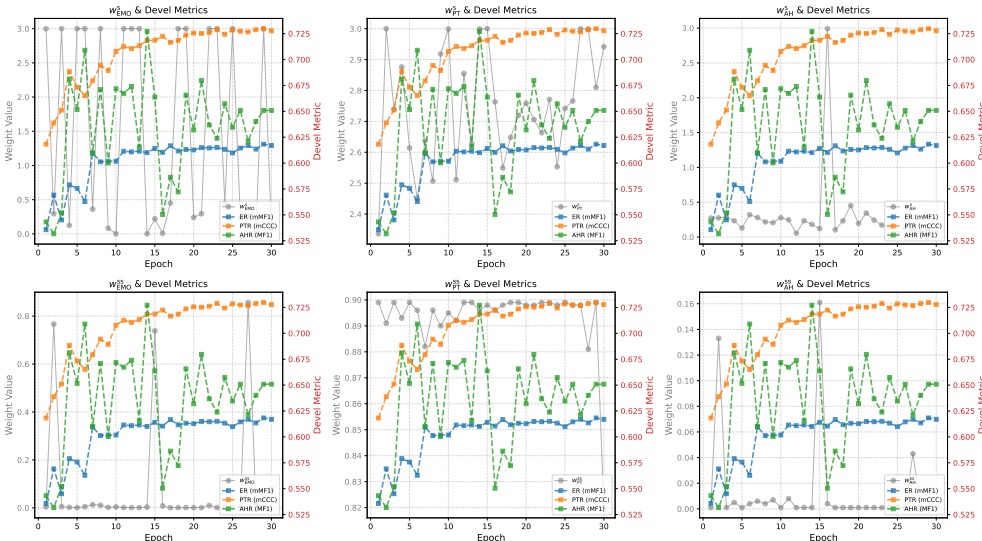

Figure 7: Visualization of adaptive change of task contribution coefficients by the best SSL hyperparameters.

training epoch using the double-branch GradNorm method. The coefficient trajectories are plotted in gray. The blue line shows the evolution of the mMF1 measure for ER, the orange line tracks mCCC for PTR, and the green line represents MF1 performance for AHR. Overall, the measures for emotion and PTs show a monotonic increase from the beginning to the end of learning, indicating stable and consistent learning. In contrast, the ambivalence curve has high volatility and is sensitive to changes in task weights. Interestingly, the weights for ambivalence are consistently low, both in supervised and semi-supervised settings. Conversely, the weights for PTs remain moderate to high with supervised learning and consistently high with SSL, suggesting that using pseudo-labels is critical for optimizing this task. The supervised weights are highly unstable for ER, while the semi-supervised weights remain persistently low. Despite their low magnitude, the semi-supervised weights for emotions and ambivalence were deliberately increased at epoch 15, coinciding with a reduction in the supervised weights. This adjustment yielded peak overall multitask performance, suggesting that strategic rebalancing towards SSL could positively impact the model's generalizability across all tasks. Even for tasks with noisy or sparse pseudo-labels, the double-branch GradNorm method can mitigate overfitting to limited labeled data and promote cross-task regularization through shared representation learning.

## A.9 CORRELATION BETWEEN TASKS AND ERROR ANALYSIS

Figure 8 illustrates the complex interplay between emotions, PTs, and ambivalence in a correlation matrix. Ambivalence shows positive correlations with negative emotions, particularly Sadness, Fear, and Disgust, and a negative correlation with Happiness, suggesting that ambivalent states are more likely to co-occur with distress-related affect rather than positive emotional experiences. No substantial correlations were observed between ambivalence and PTs, likely because ambivalence reflects

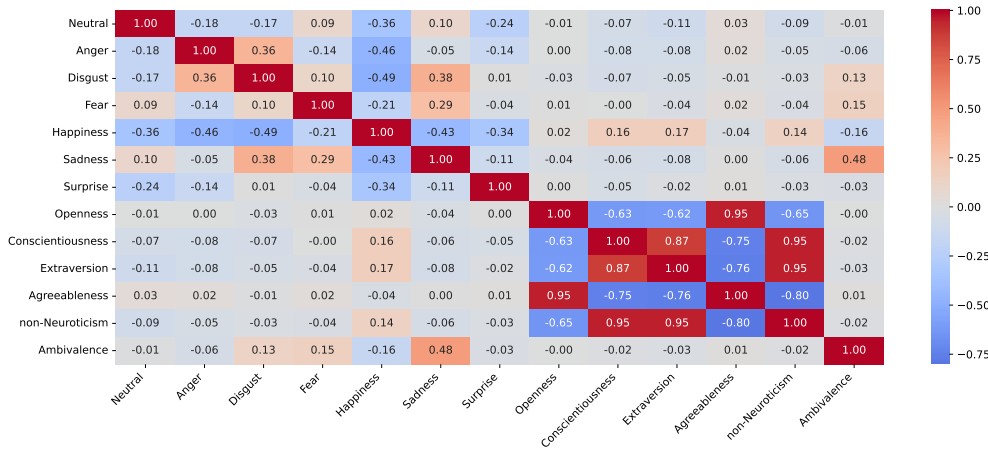

Figure 8: Visualization of the correlation between three target tasks.

a transient affective conflict rather than a stable dispositional characteristic. PTs exhibit strong positive correlations between Openness and Agreeableness and negative correlations among other Big Five dimensions. Regarding emotions, Happiness is negatively associated with most other emotional states but positively linked to Extraversion, Non-neuroticism, and Conscientiousness, aligning with established affect-trait relationships. Anger and Disgust are highly positively correlated, possibly due to overlapping expressive or semantic features in the underlying data. Disgust, Fear, and Sadness also correlate, potentially reflecting shared arousal dimensions or contextual triggers. These patterns suggest that while PTs traits form a stable, interrelated system, emotional experiences, particularly negative ones, are more dynamically intertwined with ambivalence.

Figure 9 shows the confusion matrices on MELD for different model configurations. The error analysis across the three configurations highlights the central role of learning strategies in addressing model cross-dataset generalizability to new data. By the SCD-MMPSR w/o SSL and multitask configuration, where the model was trained solely in a supervised manner on the single corpus, predictions are heavily biased toward the majority class Happiness, with more than 50% of all samples misclassified as such. This outcome reflects the uneven distribution of the training data and demonstrates the model's limited ability to generalize to less common emotion categories when faced with unseen data. Emotions such as Fear and Disgust are underrepresented in the model. The model often confuses Disgust with other emotions, such as Anger and Happiness, and Fear is confused with all emotions except Disgust.

The introduction of SSL with additional unlabeled corpora substantially mitigates this bias. By leveraging pseudo-labeling, the model in the second configuration (SCD-MMPSR w/o multitask) exhibits a more balanced distribution of predictions across emotion classes. While Happiness still dominates, the recall for Anger, Sadness, and Surprise improves, suggesting that exposure to a broader range of inputs encourages more nuanced decision boundaries. In the case of Disgust, the confusion between classes is reduced, with errors now primarily occurring in Anger and Happiness.

The full configuration (SCD-MMPSR) achieves the most consistent improvements. Incorporating auxiliary tasks (PTR and AHR) alongside pseudo-labeling introduces inductive biases, leading to significant improvements in UAR. This setup reduces the over-prediction of Happiness and strengthens recognition of Neutral, Anger, and Surprise, which benefit from richer contextual embeddings derived from the auxiliary tasks. The improved balance of classification across categories demonstrates that multitask signals help the model disentangle subtle affective cues that are otherwise obscured when optimizing for ER alone. Moreover, the problem with Fear and Disgust has been notably reduced: while in the previous two configurations both classes were predominantly misclassified as Happiness, which has the opposite valence, the errors are now redirected toward Anger, a category with closer semantic relations and overlapping multimodal patterns.

In summary, the main challenge remains a reliable minority ER, which is still affected by class imbalance and cross-domain discrepancies. Semi-supervised, cross-domain, and multitask learning

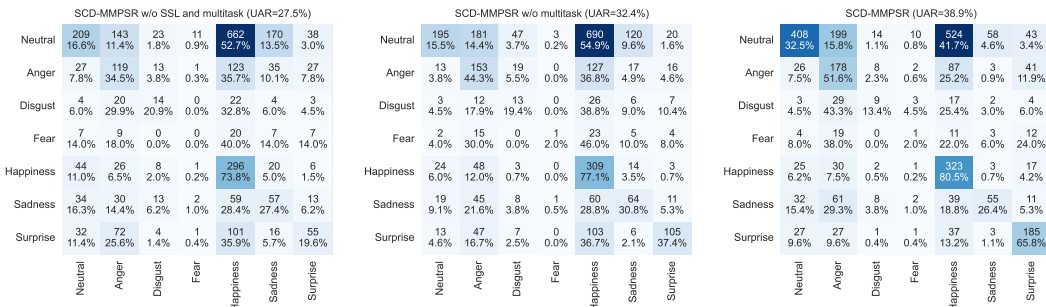

Figure 9: Confusion matrices obtained for the Test subset of the Meld corpus with different model configurations: SCD-MMPSR w/o SSL and multitask (left sub-figure), SCD-MMPSR w/o multitask (central sub-figure), SCD-MMPSR (right).

methods reduce bias and improve overall balance. However, confusion persists for semantically related categories. Future work could address these issues by adapting to the domain to align feature distributions across corpora. Combined with targeted data augmentation or reweighting strategies, this could help strengthen minority-class representations.

## A.10 COMPUTATIONAL COST

Table 17 provides a summary of the computational costs of the full MCDM model with UCGNN and MHA and its variants. The complete model incurs a moderate overhead of 9.76 M parameters, 37.2 MB in model size, and 158 seconds per epoch, primarily due to the presence of task-specific projections, graph- and cross-attention mechanisms.

Removing cross-attention layers leads to the largest reduction in model size (6.60 M parameters), while eliminating graph-attention layers results in the lowest training time (55 seconds per epoch), reflecting their computational intensity. Removing the guide bank layers slightly reduces training time, but has a negligible impact on the number of parameters or model size, confirming its lightweight design. Removing any single modality or task reduces the complexity of the model.

Table 17: Computational cost of various SCD-MMPSR configurations

| Configuration | Number of parameters, M | Model weight, MB | Learning time per epoch, s |
|---|---|---|---|
| SCD-MMPSR | 9.756 | 37.2 | 158 |
| w/o Task-Specific Projectors | 8.159 | 31.1 | 73 |
| w/o Graph Layers | 8.703 | 33.2 | 55 |
| w/o Attention Layers | 6.604 | 25.2 | 125 |
| w/o Guide Bank Layers | 9.749 | 37.2 | 147 |
| w/o Video Modality | 8.967 | 34.2 | 148 |
| w/o Audio Modality | 8.705 | 33.2 | 148 |
| w/o Text Modality | 8.705 | 33.2 | 148 |
| w/o Behavior Modality | 8.705 | 33.2 | 152 |
| w/o ER task | 7.901 | 30.1 | 111 |
| w/o PTR task | 7.905 | 30.1 | 112 |
| w/o AHR task | 7.911 | 30.2 | 120 |

It should be noted that the figures in Table 17 pertain exclusively to the SCD-MMPSR core model, trained on pre-extracted unimodal features and thus excluding the upstream feature extraction pipeline. The full framework, however, incorporates pretrained encoders for all modalities: MediaPipe for face detection (1 MB), CLIP for visual features (605 MB), CLAP for audio (615 MB), EmoRoBERTa for text (329 MB), Whisper-Turbo for speech transcription (1.62 GB), and Qwen2.5-VL-3b for behavior description generation (3.98 GB). While these components account for the majority of the total memory cost (approximately 7 GB combined), they are only used during preprocessing. When executed end-to-end on an NVIDIA A100 GPU, the complete system processes a 1-second video in 1.11 seconds on average, with 0.69 seconds spent on behavior description generation via Qwen2.5-VL-3b, highlighting it as the current computational bottleneck.

To reduce the computational cost associated with LLMs, we plan to explore several techniques in our future research. These include distilled student models, dynamic batching, caching of behavioral descriptions for repeated contexts, and more compact prompting schemes. This allows the proposed framework to provide near-real-time inference.

