# OpenReview forum: "SCD-MMPSR: Semi-Supervised Cross-Domain Learning Framework for Multitask Multimodal Psychological States Recognition"
_ICLR.cc/2026/Conference — ICLR 2026 Conference Withdrawn Submission_

### Official Review · Reviewer_AkdB · 2025-10-25

**Soundness:** 2
**Presentation:** 2
**Contribution:** 2
**Rating:** 2
**Confidence:** 4

**Summary:**

This paper focuses on psychological state recognition and proposes a joint framework, SCDMMPSR, for diverse psychological prediction tasks. The framework is characterized by semi-supervised learning, cross-domain adaptation, multi-task learning, and multimodal fusion, primarily consisting of two key components: task-specific projectors and a guide bank.

**Strengths:**

1.	A unified framework for psychological state recognition, capable of handling multiple prediction tasks simultaneously.

2.	Performance improvements across four psychological prediction tasks.

3.	Extensive experimental evaluation.

**Weaknesses:**

1.	The paper appears incremental, as it combines well-established techniques—semi-supervised learning, cross-domain adaptation, multi-task learning, and multimodal fusion. From my perspective, the main contribution is integrating these existing methods into a single framework for psychological state recognition, rather than proposing novel mechanisms.

2.	The proposed framework, as illustrated in Figure 1, seems to incrementally combine different features for multi-task learning across three tasks, without demonstrating a clearly innovative architecture.

3.	Although the framework employs complex architectures, Table 1 shows that the individual modules contribute only limited performance improvements, raising questions about their necessity and effectiveness.

4.	As shown in Table 2, the proposed method sometimes achieves only marginal gains—or even performance degradation—compared to existing state-of-the-art methods.

5.	The paper does not discuss computational efficiency or resource requirements, which is crucial for real-world applicability.

6.	Given the limited performance advantages, the paper should include statistical significance tests to ensure that observed improvements are not due to random variance.

**Questions:**

See weakness

---

> ### Author Response · Authors · 2025-11-26
> **Official Comment by Authors on Weaknesses 1-3**
>
> **Weakness 1:** The paper appears incremental, as it combines well-established techniques—semi-supervised learning, cross-domain adaptation, multi-task learning, and multimodal fusion. From my perspective, the main contribution is integrating these existing methods into a single framework for psychological state recognition, rather than proposing novel mechanisms.
>
> **Response 1:** We thank the reviewer for valuable comments. While our framework integrates well-established techniques, its novelty lies in four purpose-built components that enable effective joint learning across heterogeneous psychological tasks. First, three Task-Specific Projectors (one per task: ER, PTR, AHR) compress the input to its task’s label dimension and expands it back to the shared hidden dimension. This component forces distillation of task-relevant semantics. The projected embeddings are then refined through multimodal task-specific Graph Attention Fusion layers. These layers perform intra-modal and intra-task message passing to generate contextualized, task-aware queries. At the same time, the original multimodal input is processed by a shared multimodal Graph Attention Fusion layer. This yields a unified key-value representation which contains general representations for all task-specific domains. Each task then applies its own Task-Specific Query-Based Cross-Attention Fusion, which aligns contextualized prediction embeddings with modality features to reinforce task-specific feature representations. Finally, predictions are regularized by the Guide Bank, a set of learnable task-specific prototypes updated per batch, via cosine similarity, which stabilizes predictions. In addition, GradNorm-based adaptive weighting balances losses across tasks with disparate label spaces and annotation densities. Our ablation study, covering 29 encoders, 10 recent attention/graph modules (2024–2025), and prompt variants, shows that performance gains are not due to backbone choices but to our components. Removing  Task-Specific Graph Attention Fusion layers alone degrades average rank from 3.25 to 13.88, the largest drop observed. Thus, SCD-MMPSR is not an incremental combination of existing methods, but a novel architecture whose efficacy stems from these task-aware, prototype-guided, and adaptively balanced mechanisms, enabling robust cross-task and cross-domain psychological state recognition.
>
> **Weakness 2:** The proposed framework, as illustrated in Figure 1, seems to incrementally combine different features for multi-task learning across three tasks, without demonstrating a clearly innovative architecture.
>
> **Response 2:** We have updated Figure 1 to explicitly highlight the key innovation: semi-supervised cross-domain learning from three disjoint, single-task corpora. Unlike prior work, our framework does not require jointly annotated data, which does not exist for multi-psychological-state recognition, and instead enables knowledge transfer across tasks and domains via semi-supervision.
>
> **Weakness 3:** Although the framework employs complex architectures, Table 1 shows that the individual modules contribute only limited performance improvements, raising questions about their necessity and effectiveness.
>
> **Response 3:** We intentionally avoid training separate, task-optimized unimodal models for three reasons: (1) it would replicate the fragmented, single-task paradigm our work aims to overcome; (2) deploying multiple models incurs prohibitive computational and maintenance overhead; and (3) it prevents exploiting cross-task interaction (e.g., emotions modulating personality traits) that only emerge in joint learning. Instead, we train a single, unified model that jointly processes four modalities (audio, video, text, behavior) across all three tasks.
>
> Therefore, Table 1 reports best-performing configurations, but our full ablation studies (Appendix A.2, A.5, and A.6) show that individual encoders, attention mechanisms, or prompts yield inconsistent gains across tasks, i.e. a best-performing encoder for emotion recognition may underperform for personality or ambivalence. This confirms that no single unimodal encoders or attention mechanisms dominates in multi-task learning. We use an average rank, which helps us determine which component replacements improve performance. Critically, the ablation in Table 1 demonstrates that removing any of our four core architectural components (Task-Specific Projectors, Graph Attention Fusion, Task-Specific Cross-Attention, or the Guide Bank) causes significant performance drops. This underscores that gains stem not from auxiliary design choices, but from our proposed architecture itself, which enables a single model to effectively fuse four modalities (audio, video, text, behavior) across three heterogeneous tasks, avoiding the inefficiency and fragmentation of training separate task-specific models.

---

> ### Author Response · Authors · 2025-11-26
> **Official Comment by Authors on Weaknesses 4-6**
>
> **Weakness 4:** As shown in Table 2, the proposed method sometimes achieves only marginal gains—or even performance degradation—compared to existing state-of-the-art methods.
>
> **Response 4:** Our method is specifically designed for multi-task, cross-domain learning across multiple corpora, which presents inherent challenges: different data scales, label distributions, annotation protocol, recording conditions, and optimal learning rates for each corpus, requiring careful tuning of model parameters and training process settings. In contrast, existing state-of-the-art methods are typically optimized on a single corpus, tuned aggressively for peak performance on that specific benchmark without reporting generalization to unseen domains. While this single-corpus optimization yields higher scores on the source corpus, it compromises cross-domain robustness. Our method prioritizes generalization, not source-domain peak performance. Crucially, on the unseen MELD corpus, our multitask semi-supervised model achieves 35.04% macro F1, outperforming single-task semi-supervised baselines (27.5%) by +7.54 percentage points. This demonstrates that our framework’s strength lies not in marginal improvements on training corpora, but in scalable, transferable affective modeling, which is the central goal of our work.
>
> **Weakness 5:** The paper does not discuss computational efficiency or resource requirements, which is crucial for real-world applicability.
>
> **Response 5:** In response, we have added a new Section A.10 COMPUTATIONAL COST) that explicitly addresses the computational cost of our framework. As summarized in table, the core SCD-MMPSR model is highly compact, requiring only 37.2 MB of storage (9.76M parameters), making it suitable for deployment in resource-constrained settings. However, the full inference pipeline also relies on off-the-shelf models for feature extraction: MediaPipe for face detection (1 MB), CLIP for visual features (605 MB), CLAP for audio (615 MB), EmoRoBERTa for text (329 MB), Whisper-Turbo for speech transcription (1.62 GB), and Qwen2.5-VL-3b for behavior description generation (3.98 GB). While these components dominate the total memory cost (about 7 GB combined), they are used only during pre-processing. On an NVIDIA A100 GPU, the end-to-end system processes a single sample in 1.11 seconds, with the majority of latency (0.69 s) attributable to behavior description generation via Qwen2.5-VL-3b. To reduce LLM-related computational costs, we will explore distilled models, dynamic batching, caching, and compact prompting to enable near-real-time inference in our future research.
>
> **Weakness 6:** Given the limited performance advantages, the paper should include statistical significance tests to ensure that observed improvements are not due to random variance.
>
> **Response 6:** Table 2 reports 95% confidence intervals for all SCD-MMPSR results, computed via bootstrap resampling (1,000 samples) following standard practice  (Tibshirani & Efron, 1993). The intervals are consistently narrow across tasks, e.g., ±0.8 for MOSEI mMF1, ±1.4 for FIv2 mCCC, and ±4.5 for BAH MF1, indicating low variance and statistically reliable improvements. This confirms the improvement is not due to random variance, but to the effectiveness of cross-domain multitask semi-supervised learning. Crucially, on the unseen MELD corpus, the +7.54% gain in macro F1 (27.5 vs. 35.04) between single-task SSL and our full multitask SSL model. The expected results on in-domain benchmarks may be limited, as our model prioritizes generalization over task-specific peak performance.

---

### Official Review · Reviewer_YJYB · 2025-10-27

**Soundness:** 2
**Presentation:** 2
**Contribution:** 3
**Rating:** 4
**Confidence:** 3

**Summary:**

This paper proposes SCD-MMPSR (Semi-supervised Cross-Domain Multimodal Pretraining with Self-Reflective Regularization),
a framework designed to address the challenges of domain shift and limited labeled data in multimodal emotion recognition.
The method integrates three main components: (1) a cross-domain multimodal pretraining strategy that learns a shared semantic space across domains,
(2) a semi-supervised pseudo-labeling mechanism that leverages unlabeled samples through confidence-based consistency learning,
and (3) a self-reflective regularization term that encourages the model to calibrate its own uncertainty and enhance robustness.

**Strengths:**

1. Well-structured and conceptually coherent.
    The proposed framework is logically organized into three complementary components: domain alignment, pseudo-label learning, and self-reflective regularization.  The overall design is intuitive and theoretically grounded in representation learning principles.

2. Strong empirical results.
    The model demonstrates consistent improvements across multiple datasets and missing-data scenarios, suggesting good generalization.  The ablation analysis further highlights the individual contribution of each component, particularly the self-reflective regularization term.

**Weaknesses:**

1. Unclear pseudo-label generation and weighting.
    The description of how pseudo-labels $\tilde{y}$ are generated and weighted by confidence lacks detail.  It is not explicit whether a teacher model, EMA updates, or thresholding strategy is used, nor how the confidence scores affect optimization.

2. Notation and presentation issues.
    Ref to questions.

3. Limited theoretical insight.
    While the empirical results are convincing, the theoretical interpretation of the self-reflective regularization is somewhat heuristic.  A more formal analysis of its learning dynamics would strengthen the contribution.

**Questions:**

1. Model abbreviation inconsistency.
    The abbreviation SCD-MMPSR is written inconsistently across the manuscript.  Please unify the model name throughout the paper for clarity and consistency.


2. Undefined measure of domain discrepancy.
    In the description of the Domain Alignment Transformer, the paper claims that it “aligns the source and target domains into a shared semantic space,” but does not specify the metric used to quantify inter-domain distance.  Please clarify whether $\mathcal{L}_{align}$ is based on MMD, KL divergence, cosine distance, or another measure.

---

> ### Author Response · Authors · 2025-11-26
> **Official Comment by Authors**
>
> **Weakness 1:** Unclear pseudo-label generation and weighting. The description of how pseudo-labels  are generated and weighted by confidence lacks detail. It is not explicit whether a teacher model, EMA updates, or thresholding strategy is used, nor how the confidence scores affect optimization.
>
> **Response W1:** We thank the reviewer for these comments. As reported in Section 2.4, pseudo-labels are generated in the same forward pass as the supervised loss, without the need for a separate teacher model or EMA parameters are used. For unlabeled samples, we compute class probabilities via softmax (for ER/AHR) or sigmoid (for PTR, treated as independent binary traits). A pseudo-label is accepted only if its confidence exceeds a task-specific threshold ($τ$): for ER and AHR, we require max probability $> τ$; for PTR, we accept logits only if they lie outside the uncertainty margin $[τ, 1−τ]$, and binarize them at 0.5. The resulting pseudo-labels are then used in the standard supervised loss (CE for ER/AHR, BCE for PTR) without additional confidence weighting, only hard thresholds determine inclusion. We intentionally use a standard self-training pseudo-labeling technique (no teacher model or EMA) because our framework relies on adaptive loss balancing via dual-branch GradNorm to control the influence of pseudo-labels. This enables stable semi-supervised learning. We agree that teacher-based methods are promising and plan to explore them in future work.
>
> **Weakness 2:** Limited theoretical insight. While the empirical results are convincing, the theoretical interpretation of the self-reflective regularization is somewhat heuristic. A more formal analysis of its learning dynamics would strengthen the contribution.
>
> **Response W2:** The term “self-reflective regularization” is not used in our paper, but the reviewer may have been referring to the dual-branch GradNorm mechanism. This mechanism dynamically adjusts the task weights based on the gradient magnitudes and relative training rates. It allows the model to effectively self-regulate its learning focus across different tasks and domains. While we currently present this mechanism empirically, we have now added a brief theoretical discussion in Section 2.4 explaining how GradNorm’s balancing objective promotes stable convergence by preventing gradient imbalance, particularly under noisy pseudo-labeling. In Appendix A.8 we also visualize how this happens at each epoch.
>
> **Question 1:** Model abbreviation inconsistency. The abbreviation SCD-MMPSR is written inconsistently across the manuscript. Please unify the model name throughout the paper for clarity and consistency.
>
> **Response Q1:**  We have carefully reviewed the entire paper and unified the model name to SCD-MMPSR throughout for consistency. We note that in Table 1, we use the abbreviation MCDM (Multimodal Cross-Domain Model) to specifically refer to the trainable core component of our framework, the architecture comprising Task-Specific Projectors, Graph Attention Fusion, Cross-Attention Fusion, and the Guide Bank. Thus, SCD-MMPSR denotes the full framework with data pre-processing, while MCDM refers to its central multimodal fusion model. We have clarified this distinction in the text to avoid confusion.
>
> **Question 2:** Undefined measure of domain discrepancy. In the description of the Domain Alignment Transformer, the paper claims that it “aligns the source and target domains into a shared semantic space,” but does not specify the metric used to quantify inter-domain distance. Please clarify whether  is based on MMD, KL divergence, cosine distance, or another measure.
>
> **Response Q2:** The terms “Domain Alignment Transformer” and “aligns domains into a shared semantic space” do not appear in our paper, which may have caused confusion. Our framework does not perform explicit domain alignment, nor does it assume knowledge of domain boundaries during training. Instead, SCD-MMPSR learns from heterogeneous, task-specific corpora in a semi-supervised multitask setting.  We used one labeled and two unlabeled corpora for each task. Crucially, the model shares three multimodal feature representations across tasks as queries and one multimodal feature representation as keys and values in the Cross-Attention Fusion Layers. Domain “alignment” emerges implicitly: by leveraging unlabeled data from multiple domains via pseudo-labeling and adaptive GradNorm weighting, the model learns more generalizable features. This is empirically validated by the +7.54% macro F1 gain on MELD (35.04% vs. 27.5%) when using multitask SSL versus single-task SSL, demonstrating improved cross-domain generalization without explicit alignment.  We agree that incorporating explicit domain adaptation mechanisms (e.g., MMD or domain adversarial training) and quantifying feature-space alignment with metrics like cosine distance or MMD is a valuable direction, and we plan to explore it in future work.

---

> > ### Comment · Reviewer_YJYB · 2025-11-27
> > **Thanks for the detailed clarifications.**
> >
> > Thanks for the detailed clarifications.
> >
> > W1 This response resolves my main concern about how pseudo-labels are produced. Please add the details of how pseudo-labels are generated in the paper.
> >
> > W2 I agree the mechanism you describe aligns more with dual-branch GradNorm-style adaptive loss balancing rather than self-reflective regularization. The added discussion and per-epoch visualization should help readers understand the learning dynamics.
> >
> > Q2 Thanks for clarifying that the paper does not perform explicit domain alignment nor use an explicit discrepancy metric.
> >
> > Overall, these responses improve clarity and should reduce ambiguity for readers;  I intend to **increase my original score**.

---

> > > ### Author Response · Authors · 2025-11-27
> > > **Thanks to reviewer YJYB for increasing the score.**
> > >
> > > Thank you very much for your thoughtful feedback and for raising your score following our clarifications. We sincerely appreciate your time and constructive engagement with our work. In the final paper, we included the detailed description of pseudo-label generation and the per-epoch visualization of adaptive loss balancing to enhance clarity and reproducibility.

---

### Official Review · Reviewer_1KMA · 2025-10-29

**Soundness:** 2
**Presentation:** 2
**Contribution:** 3
**Rating:** 4
**Confidence:** 3

**Summary:**

SCD-MMPSR introduces a unified semi-supervised framework for recognizing multiple psychological states from multimodal time series data, using heterogeneous single-task corpora. The model features adaptive task weighting and novel fusion layers (Task-Specific Projectors and Guide Banks) within a graph-attention backbone to enable robust multitask and cross-domain learning, even without jointly labeled datasets. Experiments across several benchmark corpora demonstrate gains in multitask performance and generalization, highlighting the benefits of joint semi-supervised learning

**Strengths:**

- The framework supports joint multitask learning without the need for jointly labeled data.
- Improved performance in cross-domain settings.
- Extensive ablation studies demonstrate the impact of different model components.

**Weaknesses:**

- The mathematical symbols in Figure 2 and Section 2 are inconsistent, making the formulation difficult to understand.
- The explanation of the guided bank is limited. How are task-specific embeddings extracted from it?
- How does the author derive the $r_t$ formula on page 5? Why is $L_t^{(0)}$ used for normalization?
- The method depends on many hyperparameters.

**Questions:**

- The mathematical symbols in Figure 2 and Section 2 are inconsistent, making the formulation difficult to understand.
- The explanation of the guided bank is limited. How are task-specific embeddings extracted from it?
- How does the author derive the $r_t$ formula on page 5? Why is $L_t^{(0)}$ used for normalization?
- For the current formulation, Mixture of experts (MoE) are also a good alternative. It would be good if authors comment on that.

---

> ### Author Response · Authors · 2025-11-26
> **Official Comment by Authors**
>
> **Weakness and Question 1:** The mathematical symbols in Figure 2 and Section 2 are inconsistent, making the formulation difficult to understand.
>
> **Response WQ1:** We appreciate the reviewer's comments. We have revised Figure 2 and updated Section 2.3 MULTIMODAL CROSS-DOMAIN MODEL ARCHITECTURE to ensure complete consistency between the mathematical notation in the text and the visual representation. All symbols now align precisely across the figure and equations, improving clarity and readability.
>
> **Weakness and Question 2:** The explanation of the guided bank is limited. How are task-specific embeddings extracted from it?
>
> **Response WQ2:** We have expanded the description of the Guide Bank mechanism in Section 2.3 MULTIMODAL CROSS-DOMAIN MODEL ARCHITECTURE to clarify how task-specific embeddings are used. Specifically, we now explicitly state that:
> (1) the input to the Guide Bank is the refined multimodal representation produced by the task-specific cross-attention module, i.e., the embedding just before the final prediction layer;
> (2) the Guide Bank itself contains a set of learnable class prototypes (one per output class for the given task) each represented as a vector of the same dimension as the model’s hidden states, initialized randomly and updated throughout training;
> (3) for each input sample, we compute the cosine similarity between its refined representation and each class prototype, yielding a similarity score per class; this similarity vector is then averaged with the model’s raw prediction logits to produce the final output.
> These clarifications make the Guide Bank mechanism fully transparent and reproducible.
>
> **Weakness and Question 3:** How does the author derive the $r_t$  formula on page 5? Why is $L^{(0)}_t $ used for normalization?
>
> **Response WQ3:** The definition of $r_t$ in our work follows directly from the rationale introduced in the original GradNorm framework. GradNorm aims to compare the relative learning speeds of different tasks, which requires a scale-invariant measure of how much each task’s loss has decreased since the beginning of training. To achieve this, GradNorm introduces a normalized loss ratio that compares the current loss of a task to its initial valid loss value $L^{(0)}_t $. This normalization removes the influence of heterogeneous loss scales across tasks (for example, CE vs. MAE vs. BCE). It provides a dimensionless quantity that makes the learning progress of all tasks comparable. The relative inverse training rate $r_t$ is then defined by comparing each task’s normalized loss ratio to the average ratio across all tasks, indicating whether task t is learning faster or slower than the mean learning speed. We adopt this definition without modification. Using $L^{(0)}_t $as the normalization baseline is essential because it offers a stable, task-specific reference point that reflects the initial difficulty of each task and ensures that raw losses with different magnitudes become comparable. It aligns with the motivation and formulation of the original GradNorm method, and we extend it consistently to both the supervised and semi-supervised branches in our framework.
>
> **Weakness 4:** The method depends on many hyperparameters.
>
> **Response W4:** We agree with the reviewer's observation. However, most of the hyperparameters in our framework, such as batch size, learning rate, hidden dimension, and so on are standard across most deep learning models and are not specific to our architecture. The only additional hyperparameters introduced by our method are those related to semi-supervised learning (e.g., confidence thresholds for pseudo-labeling) and multi-task balancing (e.g., GradNorm's parameters). We included a note in Section 3.3 RESULTS acknowledging this and we plan to explore automated tuning or self-supervised calibration of these task-specific parameters in future work.
>
> **Question 4:** For the current formulation, Mixture of experts (MoE) are also a good alternative. It would be good if authors comment on that.
>
> **Response Q4:**  Mixture-of-Experts (MoE) architectures route each input to a subset of specialized experts through a gating network, and are effective when tasks share the same input space and differ mainly in their predictive objectives. In our case, tasks come from three disjoint corpora with different annotation schemes and domain characteristics. Under such conditions, a gating network would primarily learn to distinguish corpora rather than tasks, leading to domain-specific routing instead of meaningful task specialization. Moreover, MoE requires consistent routing for both labeled and pseudo-labeled samples, which is difficult to guarantee in a cross-domain semi-supervised setting and undermines pseudo-label stability. For these reasons, standard MoE formulations are not directly applicable to our scenario, though MoE-based variants may be explored in future work.

---

### Official Review · Reviewer_GKmR · 2025-11-01

**Soundness:** 3
**Presentation:** 2
**Contribution:** 2
**Rating:** 6
**Confidence:** 3

**Summary:**

The paper presents SCD-MMPSR, which extends existing multitask learning approaches to jointly learn emotion recognition, personality trait recognition, and ambivalence/hesitancy recognition from three independent single-task datasets. Building on established techniques including GradNorm-based adaptive task weighting, graph attention fusion, and standard pseudo-labeling strategies. The framework contributes (1) an engineering refinement that re-embeds task predictions for iterative message passing; (2) a dual-branch extension of GradNorm separating supervised and semi-supervised optimization to handle missing labels; and (3) empirical validation showing that multitask semi-supervised learning improves zero-shot generalization. While the specific three-task combination and architectural configuration are novel, the core methodological components leverage well-established techniques from the multitask learning and semi-supervised affective computing literature.

**Strengths:**

The paper addresses a well-motivated practical problem of unifying heterogeneous single-task corpora for multitask psychological state recognition without joint annotations. The experimental work is thorough, featuring comprehensive ablation studies (17 experiments) that systematically validate each architectural component, with strong empirical results on personality traits (FIv2: 92.6% mACC) and ambivalence recognition (BAH: 73.2% WF1). The framework effectively handles heterogeneous data through adaptive task weighting and semi-supervised pseudo-labeling, demonstrating cross-domain benefits with 7.5% zero-shot improvement on MELD compared to single-task learning.

**Weaknesses:**

The core techniques mentioned here are well-established, and the contribution is primarily an engineering combination of existing methods rather than fundamental algorithmic innovation. The paper primarily compares against single-task emotion recognition models rather than other multitask frameworks like MER 2025 or MuMTAffect. A comparison with these concurrent multitask approaches would help validate whether the proposed architecture components offer unique advantages. The paper would benefit from testing how the system performs without Qwen2.5-VL (using simpler video encoders) to clarify whether the strong results stem from the proposed architectural innovations or primarily from the pre-trained model selection.

**Questions:**

1. How does the proposed framework compare to other recent multitask emotion-personality approaches? Comparisons with concurrent work could strengthen the paper (if possible).
2. Could you test the framework with simpler video encoders (instead of Qwen2.5-VL) to show that the proposed components work well independently? This would help readers understand the specific value of your design choices.

---

> ### Author Response · Authors · 2025-11-26
> **Official Comment by Authors on Weaknesses 1-2**
>
> **Weakness 1:** The core techniques mentioned here are well-established, and the contribution is primarily an engineering combination of existing methods rather than fundamental algorithmic innovation.
>
> **Response W1:**  We thank the reviewer for valuable comments. While our framework integrates well-established techniques, its novelty lies in four purpose-built components that enable effective joint learning across heterogeneous psychological tasks. First, three Task-Specific Projectors (one per task: ER, PTR, AHR) compress the input to its task’s label dimension and expand it back to the shared hidden dimension. This component forces distillation of task-relevant semantics. The projected embeddings are then refined through multimodal task-specific Graph Attention Fusion layers. These layers perform intra-modal and intra-task message passing to generate contextualized, task-aware queries. At the same time, the original multimodal input is processed by a shared multimodal Graph Attention Fusion layer. This yields a unified key-value representation which contains general representations for all task-specific domains. Each task then applies its own Task-Specific Query-Based Cross-Attention Fusion, which aligns contextualized prediction embeddings with modality features to reinforce task-specific feature representations. Finally, predictions are regularized by the Guide Bank, a set of learnable task-specific prototypes updated per batch, via cosine similarity, which stabilizes predictions. In addition, GradNorm-based adaptive weighting balances losses across tasks with disparate label spaces and annotation densities. Our ablation study, covering 29 encoders, 10 recent attention/graph modules (2024–2025), and prompt variants, shows that performance gains are not due to backbone choices but to our components. Removing  Task-Specific Graph Attention Fusion layers alone degrades average rank from 3.25 to 13.88, the largest drop observed. Thus, SCD-MMPSR is not an incremental combination of existing methods, but a novel architecture whose efficacy stems from these task-aware, prototype-guided, and adaptively balanced mechanisms, enabling robust cross-task and cross-domain psychological state recognition.
>
> **Weakness 2:** The paper primarily compares against single-task emotion recognition models rather than other multitask frameworks like MER 2025 or MuMTAffect. A comparison with these concurrent multitask approaches would help validate whether the proposed architecture components offer unique advantages.
>
> **Response W2:** MER 2025 and MuMTAffect are both affective recognition frameworks, but they operate on different formulations from ours. In MER 2025, including the semi-supervised track, all methods rely on a single benchmark with shared input domain, uniform annotation, and a common emotion label space; semi-supervision there adds unlabeled data from the same domain, rather than enabling cross-corpus or cross-task learning. MuMTAffect is a fully supervised model trained on one corpus jointly annotated for both emotions and personality. Our formulation, in contrast, requires learning across three heterogeneous corpora with non-aligned label sets (ER, PTR, AHR), missing-task annotations by design, and pseudo-label propagation between corpora. This cross-domain, multitask SSL setup introduces challenges not addressed by the above frameworks. Because of these structural differences, existing emotion-personality multitask models cannot be directly applied or fairly compared without substantial re-formulation. Nonetheless, exploring connections under harmonized conditions remains a promising direction for future work.

---

> ### Author Response · Authors · 2025-11-26
> **Official Comment by Authors on Weakness 3 and Questions 1-2**
>
> **Weakness 3 and Question 2:** The paper would benefit from testing how the system performs without Qwen2.5-VL (using simpler video encoders) to clarify whether the strong results stem from the proposed architectural innovations or primarily from the pre-trained model selection.
>
> Could you test the framework with simpler video encoders (instead of Qwen2.5-VL) to show that the proposed components work well independently? This would help readers understand the specific value of your design choices.
>
> **Response W3 and Q2:** While Qwen2.5-VL generates textual behavioral descriptions (a language-mediated representation), simpler video encoders produce frame-level visual embeddings, which constitute a fundamentally different modeling paradigm. To directly address this concern, we conducted a comparison between LLM-generated behavior descriptions and scene-level visual encodings (using CLIP, ViT, ResNet-50, and DINOv2) — reported in Appendix A.5 COMPARATIVE ANALYSIS OF ENCODER PERFORMANCE (Table 11). The results show that scene-based visual encoders consistently outperform the text-based variant, suggesting that our strong performance is not solely due to the LLM. Nevertheless, we retain the LLM-based branch because it offers semantic interpretability, natural-language grounding, and compatibility with downstream reasoning modules, capabilities that latent visual features inherently lack. Our framework thus explores a complementary direction: not just what performs best today, but how to build systems that are explainable and aligned with human communication.
>
> **Question 1:** How does the proposed framework compare to other recent multitask emotion-personality approaches? Comparisons with concurrent work could strengthen the paper (if possible).
>
> **Response Q1:** While recent multitask methods, such as MuMTAffect, jointly address emotion and personality recognition using small corpora that are annotated for both tasks and recorded in laboratory conditions, our method differs in that it combines three large-scale task-specific corpora recorded in-the-wild setting, enabling cross-task knowledge transfer without the need for co-annotation. Additionally, we add an Ambivalence/Hesitancy Recognition task. In general, it allows for a more comprehensive analysis of the individual and provides a fundamentally different experimental setup.

---

### Official Review · Reviewer_rfZp · 2025-11-03

**Soundness:** 3
**Presentation:** 3
**Contribution:** 2
**Rating:** 4
**Confidence:** 3

**Summary:**

This paper focuses on a key limitation in multimodal affective computing, the lack of generalizable models capable of jointly learning across heterogeneous datasets and tasks. To overcome this, the authors propose SCD-MMPSR, a semi-supervised, cross-domain, multitask learning framework for multimodal psychological state recognition. The approach builds on a graph-attention backbone with two proposed components: (1) Task-Specific Projectors (TSPs), which map shared multimodal embeddings into task-conditioned logits and re-embed them into a unified latent space for iterative graph message passing; and (2) a Guide Bank, a learnable set of task-specific semantic prototypes that act as structured anchors to stabilize multitask learning and improve cross-domain generalization. The method is evaluated on three benchmark datasets, MOSEI for emotion recognition, FIv2 for personality trait inference, and BAH for ambivalence/hesitancy detection, and tested for out-of-domain generalization on MELD, demonstrating consistent and substantial performance gains.

**Strengths:**

1. Addresses the increasingly relevant challenge of cross-domain and multitask generalization for psychological and affective state recognition, which has strong implications for real-world HCI and social signal processing.
2. The integration of GradNorm-based adaptive task balancing with semi-supervised graph attention is a creative step forward in harmonizing heterogeneous multimodal corpora.
3. The Task-Specific Projector (TSP) design provides a principled way to handle both shared and task-specific representational subspaces.
4. The Guide Bank is conceptually interesting and intuitively appealing, using semantic prototypes as regularizers to enforce stable and interpretable multitask representations.
5. The writing is generally clear and logically structured, with good justification for each design choice.

**Weaknesses:**

1. While the Guide Bank is conceptually intuitive, the paper would benefit from more theoretical analysis or visualization showing how these prototypes evolve and influence latent space geometry. It would also be useful to contrast the proposed methods, specifically the use of the Guide Bank, against knowledge-infusion methods proposed in the literature.
2. The paper mentions “unifying heterogeneous corpora” but does not fully clarify how label space inconsistencies (e.g., emotion vs. personality) are reconciled beyond shared latent embedding alignment.
3. The generalizability is a difficult hypothesis to prove with empirical analysis. While the domain-specific focus makes sense and the experiments look reasonable, the broader generalizability aspect on any modality and any task is hard to justify.
5. While MELD is a good unseen test set, inclusion of more diverse or real-world datasets (e.g., MAHNOB-HCI, IEMOCAP) would better demonstrate scalability

**Questions:**

1. How are task label conflicts (e.g., continuous vs. categorical outputs) handled during multitask training?
2. Does the Guide Bank act as a static memory or is it dynamically updated during training? How is this different from knowledge-infusion techniques.?
3. How sensitive is the method to imbalanced data or differing annotation densities across tasks?
4. Could the architecture support few-shot adaptation to entirely new psychological tasks?

---

> ### Author Response · Authors · 2025-11-26
> **Official Comment by Authors Weaknesses 1-3 and Questions 1-2**
>
> **Weakness 1 and Question 2:** While the Guide Bank is conceptually intuitive, the paper would benefit from more theoretical analysis or visualization showing how these prototypes evolve and influence latent space geometry. It would also be useful to contrast the proposed methods, specifically the use of the Guide Bank, against knowledge-infusion methods proposed in the literature.
>
> Does the Guide Bank act as a static memory or is it dynamically updated during training? How is this different from knowledge-infusion techniques.?
>
> **Response W1:** We appreciate the reviewer's comment. Regarding related work, our Guide Bank differs from knowledge-infusion methods (e.g., methods that inject lexical ontologies or LLM-derived semantic priors) in that it learns task-specific prototypes dynamically, updated during training, from data alone, without external knowledge sources. Unlike prototype networks that compute class means from samples, our prototypes are independent, trainable parameters, enabling stable regularization even under label scarcity or noisy pseudo-labels. This design makes the Guide Bank particularly suited for semi-supervised, cross-corpus settings where external knowledge may not align with annotation protocols. We have expanded the description of the Guide Bank in Section 2.3 MULTIMODAL CROSS-DOMAIN MODEL ARCHITECTURE, while we leave detailed geometric analysis for future work.
>
> **Weakness 2 and Question 1:** The paper mentions “unifying heterogeneous corpora” but does not fully clarify how label space inconsistencies (e.g., emotion vs. personality) are reconciled beyond shared latent embedding alignment.
>
> How are task label conflicts (e.g., continuous vs. categorical outputs) handled during multitask training?
>
> **Response W2:**  Our framework does not directly reconcile label spaces - indeed, emotion (categorical), personality (continuous [0,1]), and ambivalence (categorical) have inherently incompatible output spaces. Instead, SCD-MMPSR unifies corpora at the level of representation. All tasks share the same multimodal encoder and cross-attention context (keys/values), but each maintains task-specific projectors, prediction heads, and Guide Bank prototypes. This design allows the model to leverage cross-task signals in a shared latent space while respecting label semantics through decoupled output layers. Therefore, "unification" refers to joint training on heterogeneous corpora through shared feature learning - not label space alignment. This is empirically validated by the +7.54% macro F1 gain on MELD (35.04% vs. 27.5%) when using multitask SSL versus single-task SSL, demonstrating improved cross-domain generalization.
>
> **Weakness 3:** The generalizability is a difficult hypothesis to prove with empirical analysis. While the domain-specific focus makes sense and the experiments look reasonable, the broader generalizability aspect on any modality and any task is hard to justify.
>
> **Response W3:** We agree with the reviewer that broad generalizability across arbitrary tasks and modalities cannot be fully established. We would like to clarify that in our work, "generalizability" specifically refers to the ability of our model to perform well on unseen data from a new dataset. We added “cross-dataset” term to highlight that we only focus on this particular form of generalizability. To do it, we report the results of three different training scenarios tested on the unseen MELD corpus. Training only on the labeled MOSEI dataset (single-task supervised learning). The model achieves a macro F1 score of 22.8%. Using the labeled MOSEI, unlabeled FIv2, and BAH corpora(single-task with self-supervised learning). The model improves to 27.5%. Using all three corpora (labeled MOSEI, unlabeled FIv2, and BAH) and two additional tasks (multitask with self-supervised learning), the model achieves a remarkable 35% macro F1, demonstrating a significant improvement in generalizability. These results confirm our hypothesis that our model can generalize well to unseen data from new corpora when trained on multiple tasks in a self-supervised manner.

---

> ### Author Response · Authors · 2025-11-26
> **Official Comment by Authors Weakness 4 and Questions 3-4**
>
> **Weakness 4:** While MELD is a good unseen test set, inclusion of more diverse or real-world datasets (e.g., MAHNOB-HCI, IEMOCAP) would better demonstrate scalability
>
> **Response W4:** We provide a comprehensive comparison of multimodal corpora in Section A.3 COMPARISON OF EXISTING MULTIMODAL CORPORA, where we analyze recording conditions, speech type, scale, target tasks, annotation protocol, and availability. For this research, the MELD corpus is selected for its generalization capability because it contains more naturalistic conversational scenarios compared to laboratory-recorded datasets (unlike IEMOCAP and MAHNOB-HCI), encompasses all relevant modalities (unlike AffWild2), and shares the same emotional classes as MOSEI (unlike IEMOCAP and MAHNOB-HCI). However, we plan to consider other corpora in our future research.
>
> **Question 3:** How sensitive is the method to imbalanced data or differing annotation densities across tasks?
>
> **Response Q3:** Our framework explicitly addresses cross-task data imbalance through dual-branch GradNorm, which dynamically adjusts task weights based on gradient magnitudes and relative training progress. This prevents tasks with larger or denser annotations (e.g., MOSEI) from dominating the optimization, ensuring that smaller tasks (e.g., BAH and FIv2) still contribute meaningfully. As shown in ablation studies (Table 1), removing GradNorm leads to significant performance degradation on low-resource tasks, confirming its role in balancing heterogeneous supervision. To address class imbalances, class weighting can be applied in the loss function. We also applied class weighting to the emotion recognition task; however, the results were lower than those obtained when weighting was not used. This suggests that adaptive weighting of loss contributions and the use of guide banks can help address the class imbalance problem.
>
> **Question 4:** Could the architecture support few-shot adaptation to entirely new psychological tasks?
>
> **Response Q4:** While our framework is designed to be easily extensible to new psychological tasks, it is not intended for few-shot adaptation. As the smallest labeled training dataset currently consists of approximately 1,000 examples (the BAH corpus), we recommend at least 1,000 labeled samples per task to train a reliable model for a new task. That said, SCD-MMPSR is an open and modular framework. To add a new task, researchers only need to: (i) structure their dataset similarly to existing corpora such as MOSEI, FIv2, and BAH; (ii) specify whether the task is a classification or regression problem; (iii) provide a corresponding loss function and prediction head. No other architectural changes are necessary. This design reduces the barrier to multi-task and cross-domain modeling, but it does not eliminate the need for adequate task-specific supervision.

---

### Official Review · Reviewer_ADDV · 2025-11-05

**Soundness:** 3
**Presentation:** 2
**Contribution:** 3
**Rating:** 6
**Confidence:** 3

**Summary:**

This paper introduces SCD-MMPSR, a framework for joint recognition of multiple psychological states (emotions, personality traits, and ambivalence/hesitancy) from multimodal data. The key innovation lies in training across heterogeneous, single-task corpora using semi-supervised learning with adaptive task weighting via an extended GradNorm method. The architecture features Task-Specific Projectors for iterative refinement and Guide Banks as learnable semantic prototypes. Experiments on MOSEI, FIv2, and BAH datasets, with generalization testing on MELD, demonstrate improved cross-domain performance, particularly a 7.5% macro F1-score improvement on MELD compared to single-task SSL.

**Strengths:**

1. The fragmentation of psychological state recognition across single-task, single-corpus methods represents a real barrier to deploying general-purpose affective AI systems, and the solution of leveraging heterogeneous corpora without requiring joint annotations is economically sensible and practically valuable.
2. The dual-branch GradNorm extension with delayed initialization, Task-Specific Projectors enabling iterative refinement, and the overall integration of semi-supervised cross-domain multitask learning represents a first-of-its-kind contribution to psychological states recognition.
3. Systematic evaluation of 20+ encoders across four modalities, comprehensive ablation studies with Friedman ranking, bootstrap confidence intervals, and extensive hyperparameter optimization demonstrate methodological maturity and enable fair comparison.
4. The 7.5% macro F1 improvement on MELD (35.0 vs. 27.5) convincingly demonstrates the framework's core value proposition, with error analysis showing meaningful reductions in majority class bias and more semantically coherent confusion patterns.

**Weaknesses:**

1. While the empirical results are strong, the paper would benefit from explaining why this specific combination of graph attention, task projectors, and guide banks is theoretically motivated. The Guide Bank formulation (Equation 13) particularly seems somewhat ad-hoc-why apply sigmoid transformation only to personality traits?
2. The real-time factor of 1.11s per second (0.69s from Qwen2.5-VL alone) limits practical deployment. While the authors suggest omitting behavior modality, this somewhat undermines the multimodal argument. It would be valuable to see training times, memory requirements, and computational comparisons with baseline methods.
3. MELD is a useful unseen test for ER, but it only assesses emotion transfer. For stronger “cross-domain” claims, evaluate transfer for PTR and AHR to different corpora (if available) or simulate domain shift (recording conditions, speaker distribution) and report robustness.
4. Reliance on Qwen2.5-VL for behavior descriptions dominates inference (0.69s per second) - that’s a hard barrier for real-time applications. The authors acknowledge this but don’t provide an evaluated lightweight alternative. Add experiments with a smaller VLM (distilled model) or ablate behavior modality thoroughly to quantify pragmatic tradeoffs.
5. The related work section would be stronger with citations to foundational semi-supervised methods such as FixMatch, MixMatch, and Mean Teacher; influential studies in cross-domain affective modeling like Zadeh et al. (2018) and Li et al. (2021); and key multimodal fusion research including Tsai et al. (2019) and Hazarika et al. (2020). In addition, incorporating recent advances such as LensLLM, which links large language models with affective perception, would provide a more complete and up-to-date context for the paper’s contributions.

**Questions:**

Please see the details in the weakness for the respective questions.
1. Can the authors clarify the theoretical motivation behind combining graph attention, task-specific projectors, and guide banks? Specifically, why does the Guide Bank formulation (Equation 13) apply a sigmoid transformation only to personality traits, and is there a principled justification for that choice?
2. The reported real-time factor of 1.11s per second (with 0.69s from Qwen2.5-VL alone) poses challenges for deployment. Could the authors provide details on training time, memory footprint, and computational cost compared to baseline methods, and discuss practical tradeoffs?
3. Since MELD only tests emotion transfer, how do the authors plan to support broader cross-domain generalization claims? Have they considered evaluating PTR and AHR transfer to new corpora or simulating domain shifts (e.g., recording conditions, speaker demographics) to test robustness?
4. Given the heavy reliance on Qwen2.5-VL for behavioral descriptors, could the authors test a lighter or distilled vision-language model and report how performance and efficiency change? Alternatively, could they provide a detailed ablation of the behavior modality’s contribution?
5. May consider strengthening the related work section by adding the references.

---

> ### Author Response · Authors · 2025-11-26
> **Official Comment by Authors on Weaknesses and Questions 1-2**
>
> **Weakness and Question 1:** While the empirical results are strong, the paper would benefit from explaining why this specific combination of graph attention, task projectors, and guide banks is theoretically motivated. The Guide Bank formulation (Equation 13) particularly seems somewhat ad-hoc-why apply sigmoid transformation only to personality traits?
>
> Can the authors clarify the theoretical motivation behind combining graph attention, task-specific projectors, and guide banks? Specifically, why does the Guide Bank formulation (Equation 13) apply a sigmoid transformation only to personality traits, and is there a principled justification for that choice?
>
> **Response WQ1:** We thank the reviewer for all comments. In Section 2.3 MULTIMODAL CROSS-DOMAIN MODEL ARCHITECTURE, we have expanded the theoretical motivation for integrating these components and other components of the proposed model.
>
> As for Equation 13, the sigmoid transformation is applied exclusively to personality trait predictions because, in the FIv2 dataset, personality trait scores are annotated on a continuous scale in the range $[0, 1]$. In contrast, emotion recognition and ambivalence/hesitancy recognition are classification tasks with discrete labels, for which logits (or softmax outputs) are appropriate. Thus, the use of the sigmoid function for personality traits recognition ensures that both the head output and the Guide Bank similarity scores are mapped into the same valid output range, enabling meaningful averaging in Equation 13.
>
>
> **Weakness and Question 2:** The real-time factor of 1.11s per second (0.69s from Qwen2.5-VL alone) limits practical deployment. While the authors suggest omitting behavior modality, this somewhat undermines the multimodal argument. It would be valuable to see training times, memory requirements, and computational comparisons with baseline methods.
>
> The reported real-time factor of 1.11s per second (with 0.69s from Qwen2.5-VL alone) poses challenges for deployment. Could the authors provide details on training time, memory footprint, and computational cost compared to baseline methods, and discuss practical tradeoffs?
>
> **Response WQ2:** In response, we have added a new Section A.10 COMPUTATIONAL COST) that explicitly addresses the computational cost of our framework. As summarized in the table, the core SCD-MMPSR model is highly compact, requiring only 37.2 MB of storage (9.76M parameters), making it suitable for deployment in resource-constrained settings. However, the full inference pipeline also relies on off-the-shelf models for feature extraction: MediaPipe for face detection (1 MB), CLIP for visual features (605 MB), CLAP for audio (615 MB), EmoRoBERTa for text (329 MB), Whisper-Turbo for speech transcription (1.62 GB), and Qwen2.5-VL-3b for behavior description generation (3.98 GB). While these components account for the majority of the total memory cost (approximately 7 GB combined), they are only used during pre-processing. On an NVIDIA A100 GPU, the end-to-end system processes a single sample in 1.11 seconds, with the majority of latency (0.69 s) attributable to behavior description generation via Qwen2.5-VL-3b. To reduce LLM-related computational costs, we will explore distilled models, dynamic batching, caching, and compact prompting to enable near-real-time inference in our future research.

---

> ### Author Response · Authors · 2025-11-26
> **Official Comment by Authors on Weaknesses and Questions 3-5**
>
> **Weakness and Question 3:** MELD is a useful unseen test for ER, but it only assesses emotion transfer. For stronger “cross-domain” claims, evaluate transfer for PTR and AHR to different corpora (if available) or simulate domain shift (recording conditions, speaker distribution) and report robustness.
>
> Since MELD only tests emotion transfer, how do the authors plan to support broader cross-domain generalization claims? Have they considered evaluating PTR and AHR transfer to new corpora or simulating domain shifts (e.g., recording conditions, speaker demographics) to test robustness?
>
> **Response WQ3:** We appreciate the reviewer's comment. Performing model generalization assessment is difficult for other tasks due to the lack of corpora with a similar annotation protocol. For example, for PTR:  UDIVA dataset is based on ​​social interaction analysis mostly in games; MuPTA contains only the Russian language. For AHR task, there are no other corpora available.
>
> In addition, we provide a comprehensive comparison of multimodal corpora in Section A.3 COMPARISON OF EXISTING MULTIMODAL CORPORA, where we analyze recording conditions, speech type, scale, target tasks, annotation protocol, and availability. To evaluate cross-corpus generalization in ER, we use MELD, which provides audio, video, and text modalities and is annotated with the same seven emotions as CMU-MOSEI. This sets it apart from Aff-Wild2, which primarily consists of facial reactions to movie clips and often lacks informative audio or spoken content. IEMOCAP employs a distinct emotion label set and was recorded in controlled laboratory settings, whereas MAHNOB-HCI focuses on valence-arousal dimensions and is also based in the laboratory. For PTR, both MuPTA and UDIVA rely on self-evaluation of the BigFive traits under controlled laboratory conditions, which does not reflect the FIv2 corpus. Finally, BAH is the only multimodal corpus that targets ambivalence and hesitation, making it uniquely suitable for AHR.
>
> **Weakness and Question 4:** Reliance on Qwen2.5-VL for behavior descriptions dominates inference (0.69s per second) - that’s a hard barrier for real-time applications. The authors acknowledge this but don’t provide an evaluated lightweight alternative. Add experiments with a smaller VLM (distilled model) or ablate behavior modality thoroughly to quantify pragmatic tradeoffs.
>
> Given the heavy reliance on Qwen2.5-VL for behavioral descriptors, could the authors test a lighter or distilled vision-language model and report how performance and efficiency change? Alternatively, could they provide a detailed ablation of the behavior modality’s contribution?
>
> **Response WQ4:** While Qwen2.5-VL-3B (3B parameters) is indeed the dominant factor in inference latency (0.69 s per second of input), it is already a relatively compact variant of the Qwen-VL family. In response to this concern, we have expanded Section A.2 PROPOSED PROMPT AND EXAMPLE OF BEHAVIOR DESCRIPTION with a comparison of alternative VLLMs for behavior description generation, including Eagle2-2B (2B) and InternVL2.5-4B (4B). As shown in Table 5, the smaller Eagle2-2B underperforms substantially across all tasks, despite being faster, indicating insufficient capacity to capture subtle behavioral cues. Surprisingly, even the larger InternVL2.5-4B does not consistently outperform Qwen2.5-VL-3B, suggesting that architectural design and training data matter as much as scale. These results suggest that lighter VLLMs can reduce latency, but may also result in reduced performance. We recognize this as a current challenge and plan to explore techniques such as distillation or caching in future work.
>
>
> **Weakness and Question 5:** The related work section would be stronger with citations to foundational semi-supervised methods such as FixMatch, MixMatch, and Mean Teacher; influential studies in cross-domain affective modeling like Zadeh et al. (2018) and Li et al. (2021); and key multimodal fusion research including Tsai et al. (2019) and Hazarika et al. (2020). In addition, incorporating recent advances such as LensLLM, which links large language models with affective perception, would provide a more complete and up-to-date context for the paper’s contributions.
>
> May consider strengthening the related work section by adding the references.
>
> **Response WQ5:** We expanded Section A.1 RELATED WORK  by adding references to semi-supervised methods such as FixMatch, MixMatch, and Mean Teacher; influential studies in cross-domain affective modeling (Zadeh et al. (2018) and Li et al. (2021)); and key multimodal fusion research, including Tsai et al. (2019) and Hazarika et al. (2020). Also, we included the link to the recent advance LensLLM. We have described the methods mentioned above in the relevant section.

---

### Author Response · Authors · 2025-11-26
**General comment**

We sincerely thank all reviewers for their thoughtful comments and constructive suggestions, which have significantly improved the quality and clarity of our work. In response, we have carefully addressed every concern in the revised manuscript, with all new content highlighted in red for ease of review.

Specifically:
1) In response to Reviewer ADDV, we expanded the theoretical motivation of our architecture (Sec. 2.3), clarified the use of sigmoid in the Guide Bank (Eq. 13), added a detailed computational cost analysis (Appendix A.10), and strengthened the related work (Appendix A.1).
2) For Reviewer rfZp, we clarified the mechanism of cross-corpus unification (Response W2), provided empirical evidence of generalizability on MELD, and justified corpus selection (Appendix A.3).
3) Addressing Reviewer GKmR, we added ablation studies comparing Qwen2.5-VL with lightweight visual encoders (Appendix A.5, Table 11) and clarified the novelty of our multitask design beyond engineering integration (Response W1).
 \item For Reviewer 1KMA, we unified notation (Fig. 2, Sec. 2.3), clarified Guide Bank mechanics (Response WQ2), and explained the GradNorm formulation (Response WQ3).
4) In response to Reviewer YJYB, we detailed pseudo-labeling (Sec. 2.4), removed ambiguous terminology, and clarified that domain alignment is implicit, not explicit.
5) Finally, for Reviewer AkdB, we reinforced the architectural novelty through ablation (Table 1, Appendix A.3–A.6), provided statistical significance via confidence intervals (Table 2), added a detailed computational cost analysis (Appendix A.10), and contextualized performance gains in terms of cross-domain generalization rather than in-domain peak scores.

Remaining comments primarily concern future work or methodological preferences and do not require changes to the current manuscript. Full point-by-point responses are provided above in the style requested.

We hope these revisions have fully addressed all concerns and kindly ask the reviewers to reconsider their evaluations in light of the improvements made.

---

### Note · Authors · 2026-01-19

I have read and agree with the venue's withdrawal policy on behalf of myself and my co-authors.